# Assessing the impacts of simulated Ocean Alkalinity Enhancement on viability and growth of near-shore species of phytoplankton

Jessica L. Oberlander[1], Mackenzie E. Burke[1], Cat A. London[1], Hugh L. MacIntyre[1]

[1]Department of Oceanography, Dalhousie University, Halifax, Nova Scotia, N3H 4R2, Canada

*Correspondence to*: Jessica L. Oberlander (Jessica.Oberlander@dal.ca)

**Abstract.** Over the past 250 years, atmospheric $CO_2$ concentrations have risen steadily from 277 ppm to 405 ppm, driving global climate change. In response, new tools are being developed to remove carbon from the atmosphere using negative emission technologies (NETs), in addition to reducing man-made emissions. One proposed NET is Ocean Alkalinity Enhancement (OAE), in which artificially raising the alkalinity favours formation of bicarbonate from $CO_2$, leading to a
decrease in the partial pressure of $CO_2$ in the water. Subsequent invasion of atmospheric $CO_2$ results in net sequestration of atmospheric carbon. The aim of this study was to investigate the impact of simulated OAE, through the alteration of pH, on phytoplankton representative of the spring and fall blooms in near-shore, temperate waters. The potential impacts of OAE were assessed through 1) an analysis of prior studies investigating the effects of elevated pH on phytoplankton growth rates and 2) by experimentally assessing the potential impact of short-term (10-minute) and long-term (8 days) elevation of pH on
the viability and subsequent growth rates of two representative near-shore species of phytoplankton. Viability was assessed with a modified Serial Dilution Culture – Most Probable Number assay. Chlorophyll a fluorescence was used to test for changes in photosynthetic competence and apparent growth rates. There were no significant impacts on the viability or growth rates of the diatom *Thalassiosira pseudonana* and the prymnesiophyte *Diacronema lutheri* (formerly *Pavlova lutheri*) with short-term (10-minute) exposure to elevated pH. However, there was a significant decrease in growth rates with long-term (8 days)
exposure to elevated pH. Short-term exposure is anticipated to more closely mirror the natural systems in which land-based OAE will be implemented because of system flushing and dilution. The analysis of prior studies indicates wide variability in the growth response to elevated pH within and between taxonomic groups, with about 50% of species expected to not be impacted by the pH increase anticipated from unequilibrated mineral-based OAE. To the extent that the growth responses reflect (largely unreported) parallel reductions in DIC availability, the susceptibility may be reduced for OAE in which $CO_2$
in-gassing is not prevented.

**1 Introduction**

Climate change has become one of the most pressing problems facing us as a society, with atmospheric carbon dioxide ($CO_2$) concentrations steadily increasing over the past 250 years (Dlugokencky and Tans, 2018). This led to the signing of the Paris Agreement in 2015, with the agreed-upon goal to keep the global average increase in temperature below 2 °C (*United Nations Framework Convention on Climate Change*, 2015). It is widely acknowledged, however, that reducing emissions will not be enough to meet this goal and carbon dioxide removal (CDR) will be needed. In fact, many of the IPCC scenarios that comply with the Paris Agreement regulations require as much as 10-20 Gt of $CO_2$ removal per year (Honegger and Reiner, 2018). To achieve this ambitious removal target Negative Emissions Technologies (NETs) will be needed.

Ocean Alkalinity Enhancement (OAE) is one promising NET that involves anthropogenically raising the alkalinity of a parcel of water causing the partial pressure of the $CO_2$ in that water to decrease. This change leads to either in-gassing of $CO_2$ from the atmosphere or a reduction in out-gassing of $CO_2$ from the ocean, depending on the initial air-sea gradient (Oschlies et al., 2023). Both scenarios result in a theoretical net reduction of atmospheric $CO_2$ through storage in the form of bicarbonate ($HCO_3^-$) and carbonate ($CO_3^{2-}$) ions in the ocean (Oschlies et al., 2023). The additional carbon would be stored in the form of bicarbonate and carbonate ions, with the former (which has a residence time of c. 1,000 years in the ocean) favoured after in-gassing of $CO_2$. There are currently several different methods of OAE in development, including mineral- and electrochemical-based methods, with deployment from vessels, through preexisting outfalls, or from placement on beaches (Eisaman et al., 2023). The focus of this study is the mineral-based approach from preexisting outfalls, implementation of which is likely to occur through addition of unequilibrated hydroxide minerals ($OH^-$) to the coastal surface ocean. However, before any large-scale implementation can begin, it must be confirmed that there are no negative impacts on marine biota, including phytoplankton as they are the base of the majority of marine food webs and play a significant role in biogeochemical cycling in the ocean (Winder and Sommer, 2012).

All phytoplankton rely on ribulose-1,5,-bisphosphate carboxylase/oxygenase (Rubisco) in the Calvin cycle to assimilate $CO_2$ (Raven et al., 2017). The reaction is generally undersaturated in regards to $CO_2$ since the current concentration of $CO_2$ available for phytoplankton (*i.e.*, dissolved in the ocean) is only about 10 mmol m$^{-3}$ (Raven et al., 2017) while Rubisco's half-saturation concentration is 105 – 290 mmol m$^{-3}$ (Jordan & Ogren, 1981; Tcherkez et al., 2006; Badger & Bek, 2008; Shih et al., 2016). The reason for the low concentration of dissolved $CO_2$ usable by phytoplankton is well illustrated in a Bjerrum plot (*e.g.*, Zebee and Wolf-Gladrow, 2001) which shows bicarbonate's dominance of the inorganic carbon species at pH values of 6 – 9. Although $CO_2$ is the substrate for Rubisco, the prevalence of $HCO_3^-$ underlies a strong selective pressure among phytoplankton for the ability to utilize $CO_2$ in a carbon concentrating mechanism (CCM). This is a trait observed across taxonomic groups (Colman et al., 2002; Nimer et al., 1997; Beardall et al., 2020). Different CCMs facilitate uptake of $CO_2$ by its active transport across the cell membrane and/or by uptake of $HCO_3^-$ through anion exchange (Beardall and Raven, 2016), followed by its

conversion to $CO_2$ by carbonic anhydrase (Coleman et al., 2002l Nimer et al., 1997; Beardall et al., 2020). Taxonomic differences in the energetic costs of different CCMs (Raven et al., 2014), in the pH optima of different forms of carbonic anhydrase (Beardall et al., 1976; Idrees et al., 2017; Supuran, 2023), and in the specificity of different forms of Rubisco for $CO_2$ vs $O_2$ (Iñiguez et al., 2020) suggest that alkalization has the potential to alter community growth rates or to cause shifts in taxonomic structure within mixed assemblages.


Previous research has documented a clear relationship between phytoplankton growth rates and pH, with deviations from the norm in either direction resulting in negative impacts (see Supplement 1; Bach et al., 2015; Gately et al., 2023; Langer et al., 2006). There are, however, certain species of phytoplankton, notably the diatom *Phaeodactylum tricornutum*, that are able to maintain growth rates that, though lower than their maximum are still positive, at pH values above nine and below seven

(Berge et al., 2010; Hansen, 2002). A decrease in growth rate could be due to a change in vitality (*i.e.*, reduced metabolic competence, e.g., a reduction in photosynthetic efficiency), or because of a reduction in viability (i.e., reproductive competence) in a fraction of the population.

This study addressed potential impacts of OAE via the response of phytoplankton growth rates, viability, and photosynthetic

competence via responses to elevated pH. First, published data were fitted to a model of growth to quantify the effect of progressively rising pH on the growth rates of a range of cultured phytoplankton. Second, the viability, growth rates, and photosynthetic competence (as $F_v/F_m$) were measured for two representative near-shore phytoplankton species, the diatom *Thalassiosira pseudonana* and the prymnesiophyte *Diacronema lutheri* (formerly *Pavlova lutheri*), following exposure to short- (10 minutes) and long-term (8 days) elevated pH.

**2 Methods**

**2.1 Literature Review & Data Digitization**

A literature review was conducted to collate data from various studies that have used either batch or semi-continuous cultures to assess the effect of elevated pH on growth rates of phytoplankton. The search terms included "phytoplankton AND (alkalinization OR "high pH")". To ensure consistent approaches were used, studies found in the search were gated to include

only those in which cultures were maintained under pH drift conditions, with culture medium ensuring that the media would have DIC:DIN below the Redfield Ratio, and with time-series (growth curves) of phytoplankton abundance and pH. Additional criteria included species/strains of phytoplankton indicative of marine systems and the inclusion of irradiances and temperature in the methods. The majority of the studies meeting the above criteria were by Hansen and colleagues, (see Supplementary Table 1). Their protocols controlled nutrient stoichiometry to ensure DIC limitation of growth and most were in batch cultures

without ventilation to replenish $CO_2$. It is critical to recognize that in pH-drift cultures such as these, both pH and DIC vary

and that the biological response may be due to changes in $CO_2$ availability as much as, or more than, by pH (see Section 3.2, below). A summary is given in Supplementary Table 1.

The data from each study was digitized using OriginPro 2022b (9.9.5.167, Learning Edition) to test for the relationship between growth rate and pH. The data were either growth curves (variations in cell concentration over time, from which growth rate can be estimated) or growth rates and parallel variations in pH. To determine error associated with the digitization, random numbers were generated and plotted against a linear series and the plot was then digitized the same way. The root mean square error (RMSE) of the y-axis values in the randomly generated plot was 0.4% of the maximum of the data range.

## 2.2 Model Fitting

For articles that listed cell concentrations, these were ln-transformed and fit against time using Equation 1. This is a modification of the model used by Bannister (1979) to describe the photosynthesis-irradiance response curve, recast to include a non-zero intercept. The choice of Bannister's formulation over other commonly used growth models (Zwietering et al., 1990) was based on its flexibility in accommodating abrupt versus gradual transitions between the exponential and stationary phases of the response (Jones et al., 2014).

$$\ln[cells_t] = \left(\ln[cells_{fin}] - \ln[cells_{init}]\right) \cdot \frac{t}{\left((t_{stat})^b + t^b\right)^{\frac{1}{b}}} + \ln[cells_{init}] , \qquad (1)$$

where $t$ is time (d) and $t_{stat}$ is the time to stationary phase; $cells_{init}$, $cells_{fin}$, and $cells_t$ are cell concentrations (cells mL$^{-1}$) at $t = 0$, in stationary phase, and at time $t$; and b (dimensionless) is a parameter that defines the curvature in the function as the growth rate declines in the transition between exponential and stationary phase.

The maximum and time-dependent growth rates, $\mu_m$ and $\mu$ (d$^{-1}$), were then determined empirically from the fit to Equation 1 as tangents to the growth curve calculated in increments of 0.014 d. The growth rate was expressed in dimensionless terms as $\mu/\mu_m$ for comparison between studies. These relative growth rates were calculated at times when corresponding pH data were collected and the resulting curve was fitted with either a biphasic (Equation 2a & b) or 1$^{st}$-order kinetic model (Equation 2a & c) that describes the decline in growth rate above a threshold pH value (modified from MacIntyre et al., 2018);

$$\frac{\mu}{\mu_m} = 1 \text{ for pH} \le \text{pH}_{Th} , \qquad (2a)$$

and

$$\frac{\mu}{\mu_m} = (1 - \alpha) \cdot \exp\left(-k_1(pH - pH_{Th})\right) + \alpha \cdot \exp\left(-k_2(pH - pH_{Th})\right) \text{ for pH} > \text{pH}_{Th} , \qquad (2b)$$

or

$$\frac{\mu}{\mu_m} = \exp\left(-k \cdot (pH - pH_{Th})\right) \text{ for pH} > \text{pH}_{Th} , \qquad (2c)$$

where α (dimensionless) is a coefficient that partitions the biphasic response and varies between 0 and 1, $k_1$ and $k_2$ and $k$ (pH$^{-1}$) are sensitivity coefficients that relate to the slopes; and pH$_{Th}$ is the threshold pH above which growth rates decline with pH. The threshold model of pH change is utilized here as it is a common model that allows "shouldering" in various studies of inactivation (Hijnen, 2006; Weavers & Wickramanayake, 2001). Although the focus of these studies is UVC photoinactivation rather than pH limitation, the mechanistic underpinning of progressive debilitation, often with "tailing"

(*i.e.*, a biphasic response) is supported by observations of "shouldering" in the data followed by progressive debilitation in both scenarios.

For those studies that reported growth rates rather than cell concentrations, the fitting with Equation 1 was omitted, and the digitized data were fit only with Equation 2. The choice of the biphasic or 1$^{st}$-order model was based on the values of $k_1$ and

$k_2$ in the biphasic model. If the means differed by less than the sum of the standard errors of the estimates, the 1$^{st}$-order model was used; otherwise, the biphasic model was. Data sets that could not be fit with the models because there were fewer data points than fit parameters or because no pH threshold was defined were omitted from subsequent analysis.

The model fits were used to generate a pH-dependent growth curve in increments of 0.005 pH units to define the pH values

at which there was a 10% and 90% reduction in growth rate. Primer 7 software (PRIMER-e v7.0.21) was used for multivariate testing for differences in these values between taxonomic groups using analysis of similarity (ANOSIM).

### 2.3 Culturing Techniques

The phytoplankton cultures used for examining the impacts of simulated OAE were the diatom *Thalassiosira pseudonana*, Clone CCMP 1335, and the prymnesiophyte *Diacronema lutheri* (formerly *Pavlova lutheri*), Clone CCMP 1325, both of which

were obtained from the National Center for Marine Algae and Microbiota (NCMA, East Boothbay, ME, USA). These were chosen as they are representative of taxa that dominate during the spring and fall blooms of near-shore, temperate waters. The cultures were maintained in 40-mL volumes of sterile-filtered f/2 (Guillard, 1975) or L1 (Guillard & Hargraves, 1993) seawater medium, and diluted into fresh media in mid-exponential phase in a laminar flow hood. The seawater was collected by pump from a depth of 6 m below surface, about 100 m offshore from the National Research Council of Canada's Marine Institute at

Ketch Harbour, NS, and tangential-flow filtered on collection. It was refiltered through a 0.2-μm capsule filter (Cytiva Whatman Polycap Disposable Capsules: 75TC) and nutrient-enriched in autoclaved glassware or in sterile cell culture plates. Prior to experimentation, parent cultures were fully acclimated to the experimental growth conditions, continuous illumination at c. 190 μmol photons m$^{-2}$ s$^{-1}$ at a temperature of $18 \pm 1°C$, by maintaining them in balanced growth in semi-continuous culture (MacIntyre & Cullen, 2005).


Cultures were monitored daily via chlorophyll a fluorescence, measured with a Turner 10AU fluorometer (Turner Designs, USA) and a FIRe fluorometer following a 30-minute dark acclimation period. The fluorometers were blanked daily with culture

medium, and the FIRe was also standardized with a solution of 100 µmol L$^{-1}$ rhodamine *b* in E-Pure water. The estimates were corrected for the sensitivity setting and the blank. Single-turnover induction and relaxation curves measured with the FIRe

were fit with Fireworx software (Audrey Ciochetto, née Barnett, http://sourceforge.net/projects/fireworx/) to estimate minimum ($F_0$), maximum ($F_m$), and variable fluorescence ($F_v = F_m - F_0$), the quantum yield of electron transport at PSII, $F_v/F_m$, the photosynthetic cross-section, $\sigma$ ($Å^2$), and the turnover time of the PQ pool, $\tau$ (ms).

Daily specific growth rates, $\mu$ (d$^{-1}$) were calculated from the change in fluorescence measured with the 10AU according to Equation 3 (Wood et al., 2005):

$$\mu = \frac{\ln (F_t/F_0)}{\Delta t},$$ (3)

where $F_t$ is the fluorescence (Arb.) at the end of the time interval, $F_0$ is the fluorescence at the beginning of the time interval, and $\Delta t$ (d) is the length of the time interval. Cultures were assumed to be in balanced growth when the coefficient of variation (CV) for daily estimates of $\mu$ and the quantum yield of photosystem II (PSII) electron transport, $F_v/F_m$, were <10% over 10 generations (MacIntyre & Cullen, 2005). Experiments were initiated once a culture was in balanced growth.

**2.4 Experimental Setup**

For a summary of experimental conditions and measurements, see Table 1.

**2.4.1 Comparison of aerated and pH-drift cultures of *Thalassiosira pseudonana***

Cultures of the diatom *Thalassiosira pseudonana* (CCMP 1335) in 40-ml glass tubes were maintained in balanced growth (*i.e.*, in semicontinuous culture) in f/2 medium at 18 °C and under continuous illumination of 190 µmol photons m$^{-2}$ s$^{-1}$, provided

by cool-white fluorescent bulbs (see Section 2.3 for more details). The cultures were used to inoculate 2-L volumes of f/2 medium (Guillard, 1973). One was sealed (pH-drift); the second (aerated) was bubbled with air that had been passed through an activated charcoal filter (MacIntyre and Cullen, 2005). Both were stirred with Teflon-coated magnetic stir bars.

The two cultures were subsampled daily for analysis of DIC, pH, cell counts, and *in vivo* fluorescence. The DIC and pH were

used to estimate the concentrations of $CO_2$, bicarbonate, and carbonate, using CO2SYS (Lewis and Wallace, 1998). Cell counts were performed with an Accuri C6 flow cytometer (BD, Franklin Lakes, NJ, USA), as described by MacIntyre et al. (2018). Fluorescence induction and relaxation (FIRe) curves were measured with a FIRe fluorometer (Satlantic, Halifax, NS, Canada), to estimate $F_v/F_m$ , $\sigma$, and $\tau$.

The variation in the biological parameters was compared to the abiotic parameters using BEST (Clarke et al., 2008), an iterative test based on correlations between a matrix of pairwise similarity coefficients based on the biotic data, against similar matrices of all possible combinations of 1-5 abiotic parameters.

*Table 1*: Summary of experiment conditions and measurements in the culture experiments. Abbreviations: DIC, dissolved inorganic carbon; Chl, chlorophyll *a*; FIRe curves, chlorophyll *a* fluorescence induction and relaxation curves (measured with FIRe); *F*, chlorophyll *a* fluorescence in vivo, measured with Turner 10AU (chronic exposure) or Synergy4 plate reader (transient exposure).

| Experimental Comparison | Treatment (pH) | Exposure Duration | Culture Volume | Gas Exchange | Parameters | Measurement frequency |
|---|---|---|---|---|---|---|
| Effect of chronic alkalization ± $CO_2$ replenishment on growth rate in *T. pseudonana* | Drift | 7 days | 2-L bottles | Aerated vs sealed | DIC pH Chl Cell counts FIRe curves | Daily |
| Effect of chronic alkalization on growth rate in *T. pseudonana* & *D. lutheri* | 8.02 8.13 8.34 8.57 8.69 8.82 | 8 days | 2-L bottles | Sealed | pH, Cell counts<br><br>FIRe curves, *F* | $T_0$<br><br>Daily |
| Effect of transient alkalization on viability and growth rate in *T. pseudonana* & *D. lutheri* | 7.96 8.18 8.34 8.47 8.60 8.69 8.83 8.87 9.00 9.06 9.09 | 10-minutes (SDC – MPN)<br><br>1-2 hours (FIRe curves) | 1-mL wells in plates (SDC – MPN)<br><br>6-mL (FIRe curves) | Diffusive | pH, Cell counts, FIRe curves<br><br>*F* | $T_0$<br><br>Daily |

**2.4.2 Transient and chronic OAE effects on viability, growth, and photosynthetic competence**

All the transient and chronic exposure experiments were conducted on a light table continuously illuminated from below by white LEDs at c. 190 µmol photons $m^{-2}$ $s^{-1}$, at a temperature of $18 \pm 1°C$ (*i.e.*, the same conditions to which the cultures had been acclimated). Incubations were set up by alkalizing cultures with increasing volumes of a 0.5-mol $L^{-1}$ solution of NaOH to increase the pH and alkalinity. Treatments consisted of additions of $100 – 1000$-µmol $L^{-1}$ NaOH (final concentrations). These additions increased the initial concentration of total alkalinity, 2168 µmol $L^{-1}$, by $0 – 1084$ µmol $L^{-1}$ (All reported values

of alkalinity are calculated from measured values of pH and DIC). For the transient exposure experiments, the culture and NaOH were mixed and allowed to react for 10 minutes before dividing into two aliquots. The first was used immediately for measurement of pH and the serial dilutions. The second was used for measurement of $T_0$ fluorescence, FIRe curves ($F_v/F_m$), and cell counts. These parameters were measured following set-up of the SDC – MPN assays, 1-2 hours after alkalization. For the chronic exposure experiments, the culture was not diluted. The pH was measured immediately following manipulation with the NaOH. The $T_0$ values of $F_v/F_m$ and cell counts were measured after approximately a 30-minute dark acclimation period. The FIRe curves were measured daily thereafter by inserting the sealed culture tubes into the instrument's cuvette holder.

Viability of cultures in the transient exposure experiments was measured with the Serial Dilution Culture – Most Probable Number (SDC – MPN) assay (McCrady, 1915; Throndsen, 1978), using the methodology developed and validated by MacIntyre et al. (2018, 2019). The method was modified to use sterile 24-well polystyrene cell culture plates (Sigma Aldrich), based on the approach described by Bernd and Cook (2002), rather than individual tubes. The assays were set up with 8 replicates in each of 6 tiers of successive $10^{-1}$ dilutions and the plates were kept in sealed glass boxes. The dilutions were made with fresh sterile culture medium. The concentrations of viable cells at the end of the grow-out period were estimated using an online calculator (EPA, USA; https://mostprobablenumbercalculator.epa.gov/mpnForm). These were expressed relative to the initial cell concentration, estimated using a hemocytometer with an inverted microscope (Leica Microsystems), as described by MacIntyre et al. (2018), as the Relative Viability (dimensionless), as described by MacIntyre et al. (2018). Chlorophyll fluorescence was monitored daily with a Synergy4 plate reader (BioTek, Winooski, VT, USA) after dark acclimation for 30 minutes, Following measurement, the plates were returned to the transparent boxes. To minimize the likelihood of desiccation or carbon limitation of growth, the contained an open 100-ml container of a 1-mmol $l^{-1}$ solution of sodium bicarbonate ($NaHCO_3$Immediately before closure, 400 µL of 10% HCl was added to the $NaHCO_3$ reservoir to generate $CO_2$, minimizing the likelihood of the cultures becoming carbon limited.

The fluorescence-based growth rate of the two sets of cultures was measured in one of two ways. For the transient exposure experiments, the fluorescence of the $10^{-3}$ dilution of the SDC – MPN was measured daily with the Synergy4 plate reader. For the chronic exposure experiments, it was measured daily with a 10AU fluorometer in parallel with the FIRe curves. Cultures were grown into stationary phase and the instantaneous and maximum growth rates were calculated using Equations 4a, 4b, and 5. Equation 4 is a modification of Equation 1 that allows for a period of monitoring in which the signal does not change, either because the culture is in lag phase or because the signal is below the lower limit of detection.

$$\ln[F_t] = \ln[F_{init}] \text{ for } t \leq t_{\text{lag}} \tag{4a}$$

$$\ln[F_t] = \left(\ln[F_{fin}] - \ln[F_{init}]\right) \cdot \frac{t - t_{lag}}{((t_{exp})^b + (t - t_{lag})^b)^{\frac{1}{b}}} + \ln[F_{init}] \text{ for } t > t_{\text{lag}} \tag{4b}$$

$$\mu_{m=}\frac{\ln[F_{fin}]-\ln[F_{init}]}{t_{exp}} \hspace{3cm} (5)$$

where $t$ is time (d); $F_{init}$, $F_{fin}$, and $F_t$ are fluorescence (Arb.) at $t = 0$, in stationary phase, and at time $t$; $t_{lag}$ and $t_{exp}$ are the durations of the lag and exponential phases of growth (d); and $b$ (dimensionless) is a parameter that defines the curvature as the growth rate declines in the transition between exponential and stationary phase.

## 3 Results

### 3.1 A review of the pH dependence of growth rates

The approach to estimating the pH thresholds for changes in growth rate is illustrated in Figure 1a and b, using data reported by Berge et al. (2012). The changes in cell concentration over time (Figure 1a) were fitted to Equation 1 (Figure 1a). The variations in growth rate (Figure 1a) was derived as the tangent to the curve fit. The values of growth rate at the times pH was measured (Figure a) were then fit to the pH values using Equations 2a, 2c, and expressed as the ratio to the maximum value, $\mu_m$, (Figure 1b). The normalized growth curves for all species (see Supplementary Table 1 for sources) are shown in Figure 1c. The threshold pH and pH values at which there was a 10% and a 90% reduction in growth rate are compared in Figure 1d. The data were grouped by taxonomic status – divisions, except the classes containing raphidophytes and diatoms within the Heterokontophyta – and compared by ANOSIM. There are statistically significant difference between groups (p = 0.001), with the major differences being between dinoflagellates and all other groups except for prymnesiophytes, and between cryptophytes and diatoms (Table 2). A similarity percentage analysis (SIMPER) shows that the factor driving this difference is the pH at which $\mu$ was reduced by 90% in most cases, indicative of taxonomic differences in response to extreme alkalization. In some comparisons with dinoflagellates, the difference is more evenly distributed between the three pH factors, indicating higher sensitivity of the dinoflagellates at lower levels of alkalization.

*Table 2*: R significance level (ANOSIM) for differences between taxonomic groups in Figure 1c and d. Variables include the threshold pH, pH for a 10% reduction in growth rate, and pH for a 90% reduction in growth rate. Significant differences are indicated by asterisks.

| | Crytophyte | Prymnesiophyte | Chlorophyte | Raphidophyte | Diatom | Dinoflagellate |
|---|---|---|---|---|---|---|
| **Prymnesiophyte** | 0.2 | | | | | |
| **Chlorophyte** | 0.2 | 0.2 | | | | |
| **Raphidophyte** | 1 | 0.33 | 0.25 | | | |
| **Diatom** | 0.0071* | 1 | 0.1 | 0.14 | | |
| **Dinoflagellate** | 0.001* | 0.89 | 0.003* | 0.019* | 0.054* | |

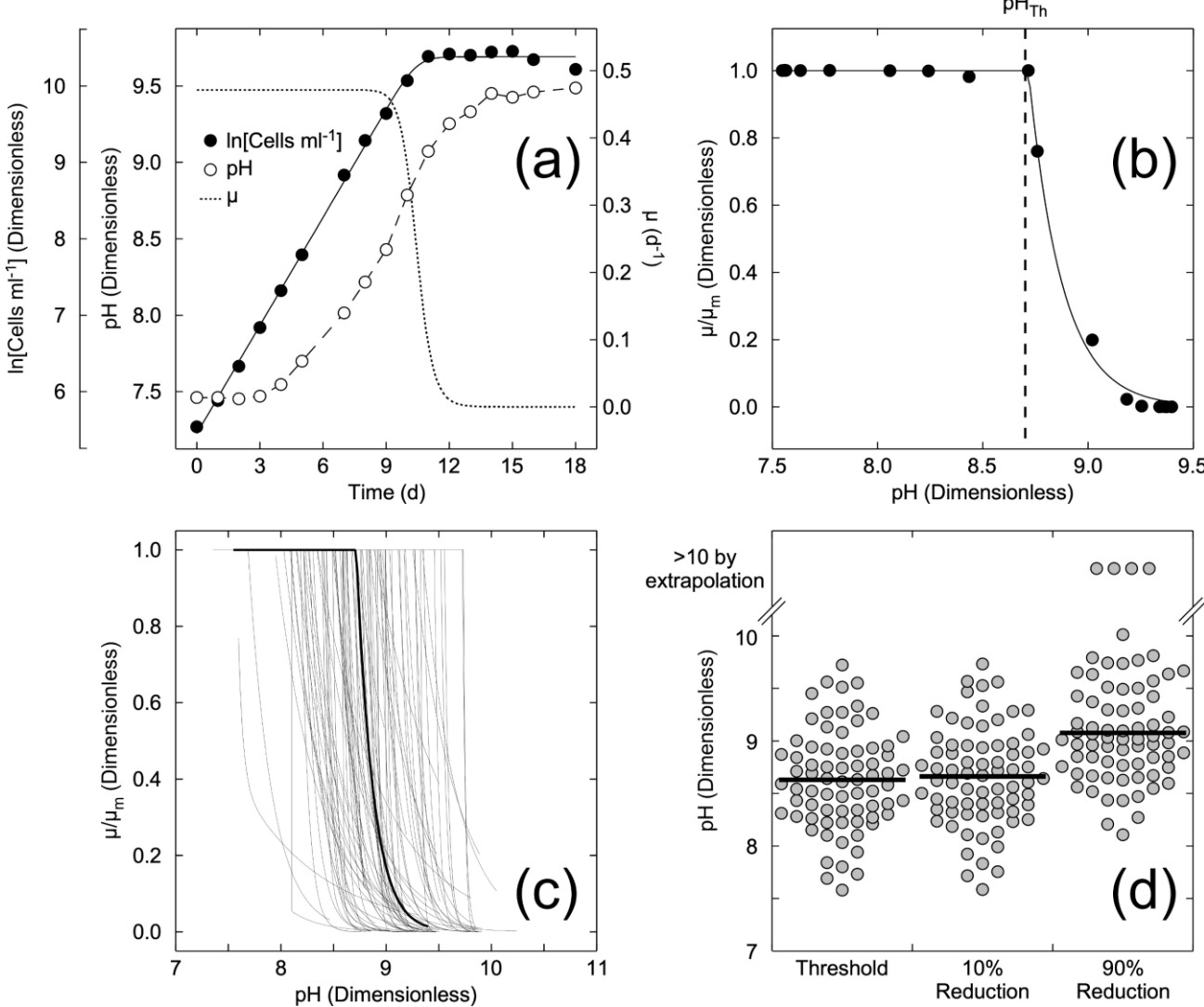

*Figure 1*: (a) Example of the Bannister model (Equation 1) fit for *Heterocapsa triquetra*, NIES 7 (Berge et al., 2012), with the modelled growth rate and corresponding pH. (b) Example of the 1st-order kinetic model fit (Equation 2a, c) for the same data. The threshold pH, $pH_{Th}$ (dashed line) was 8.72±0.005. (c) Compilation of the 72 dimensionless relationships between relative growth rate and pH for the different species or environmental conditions (see Supplementary Table 1). The black line is the fit for *Heterocapsa triquetra*, NIES 7, in panels a and b. (d) The calculated threshold pH, pH for a 10% reduction in growth rate, and pH for a 90% reduction in growth rate for the growth curves in (c). The black lines represent the median values.

For all of the species investigated (Figure 1d), the median threshold pH is above 8.5, suggesting about 50% of species would not be impacted by the anticipated maximum pH increase at the point of addition for a mineral-based addition of OAE from a land-based point source. The other 50% of species might be impacted. The variability in response could be attributed to a number of different factors including the concentrations of DIC (Hansen et al., 2007; Søderberg & Hansen, 2007; Søgaard et al., 2011), light intensity (Søderberg & Hansen, 2007; Nielsen et al., 2007), or adaptive differences among isolates. However, the effects of DIC and light intensity cannot be assessed directly from these studies as only three of the thirteen studies report DIC availability, and only two included variations in light intensity in the experimental design.

These studies also allowed a limited number of comparisons of growth rates between cultures maintained at constant, elevated pH in semi-continuous culture and those where the pH was allowed to drift in sealed batch cultures. A comparison of responses under these conditions is shown in Figure 2 for three dinoflagellates in the genus *Ceratium*. The average threshold pH in the constant-pH experiments is not statistically different than the drift experiments when results for *C. furca* and *C. fusus* in both conditions were grouped with the responses of *C. tripos* in pH-drift culture. In *C. tripos*, there was no detectable decrease in growth rate with pH in the semi-continuous cultures, although there was in the batch-drift cultures. As the absence of an effect could not be parameterized within the framework of the model, the semi-continuous culture is not presented in Figure 2.

In *C. furca* and *C. fusus*, the similarity of responses in the constant-pH and pH-drift cultures suggest that differences in other parameters between the culture methods (DIC, light, nutrients, etc.) have less effect on growth than changes in pH, and/or that there was no long-term acclimation to pH in the semi-continuous cultures. In contrast, there were significant differences in *C. tripos* between the conditions, with no observable effect of pH in the constant-pH culture. This might be because the difference is driven primarily by another factor, likely reduced DIC in the pH-drift experiment, or because the cells were able to acclimate to elevated pH after longer exposure. However, both experimental set-ups are a mismatch for the likely discharge of alkalinity into well-mixed waters where exposure time will be much shorter.

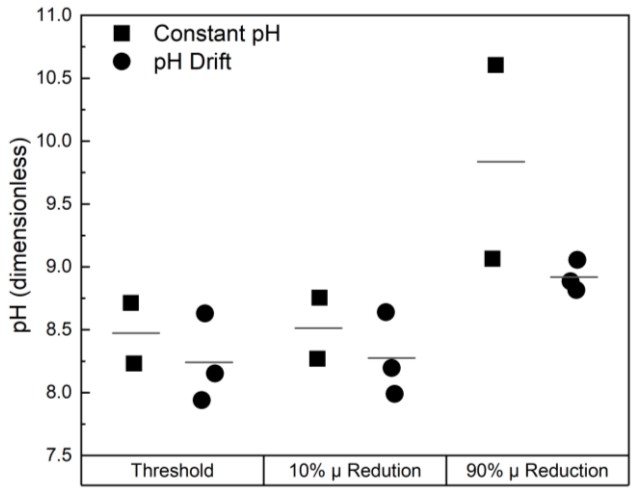

*Figure 2:* Comparison of the pH dependencies of growth in the dinoflagellates *Ceratium furca*, *C. fusus*, and *C. tripos* under two different growth conditions. Cultures were maintained at a constant pH in semi-continuous culture (Constant pH) and were kept in batch cultures in which the pH was allowed to drift (pH Drift). For each comparison, the symbols are the estimates from fits to Eq. 2 of the threshold pH, above which the growth rate declines (pHTh); the pH at which growth rate is reduced by 10%; and the pH at which growth rate is reduced by 90%. The horizontal black lines are the mean values across species under each growth condition. Growth rates in *C. tripos* did not vary over the range of pH tested when pH was kept constant, although they declined over the same range in the pH-drift experiment. Consequently, there are 2 estimates for the Constant pH condition and 3 for the pH Drift condition.

### 3.2 Examining the impact of prolonged, elevated pH on phytoplankton with and without DIC resupply

Most of the studies analyzed in Section 3.1 did not permit $CO_2$ in-gassing, as they were conducted using a closed-bottle, batch-culturing method. However, for OAE to successfully function as a NET, in-gassing of $CO_2$ is required to increase the DIC pool. The effect of this is illustrated in Figure 3a and 3b, in which batch cultures of the diatom *Thalassiosira pseudonana* Clone CCMP1335 were either aerated or not. (Details of the culture conditions and parameter estimates are given in Supplement 2). There are clear differences between the two cultures in pH, DIC, and a proxy for biomass. There

were also clear differences in the descriptors of the diatom's abundance and physiological status.

In the sealed (pH drift) incubation, there was a progressive draw-down of DIC to about 50% of the initial value and a rise in pH to $9.26 \pm 0.03$ in stationary phase (Days 4-6; Figure 3a). Estimation of the portioning of the carbonate system with CO2SYS (Lewis and Wallace, 1998) showed that the DIC pool was dominated by carbonate ($63 \pm 1\%$ of the DIC pool) and

$CO_2$ concentrations were <0.2 $\mu mol^{-1}$ in this period. In contrast, the initial draw-down of DIC was reversed in the aerated

culture when the culture entered stationary phase and $CO_2$ demand was reduced, returning to the initial value. The initial

increase in pH was significantly reversed over the same period, and average 8.53 ± 0.05 over the last 3 days of stationary

phase (Days 6-8; Figure 3b). Based on CO2SYS, the DIC pool was dominated by bicarbonate (79 ± 1%) and $CO_2$

concentrations were >3.5 $\mu mol^{-1}$ and rising during this period.

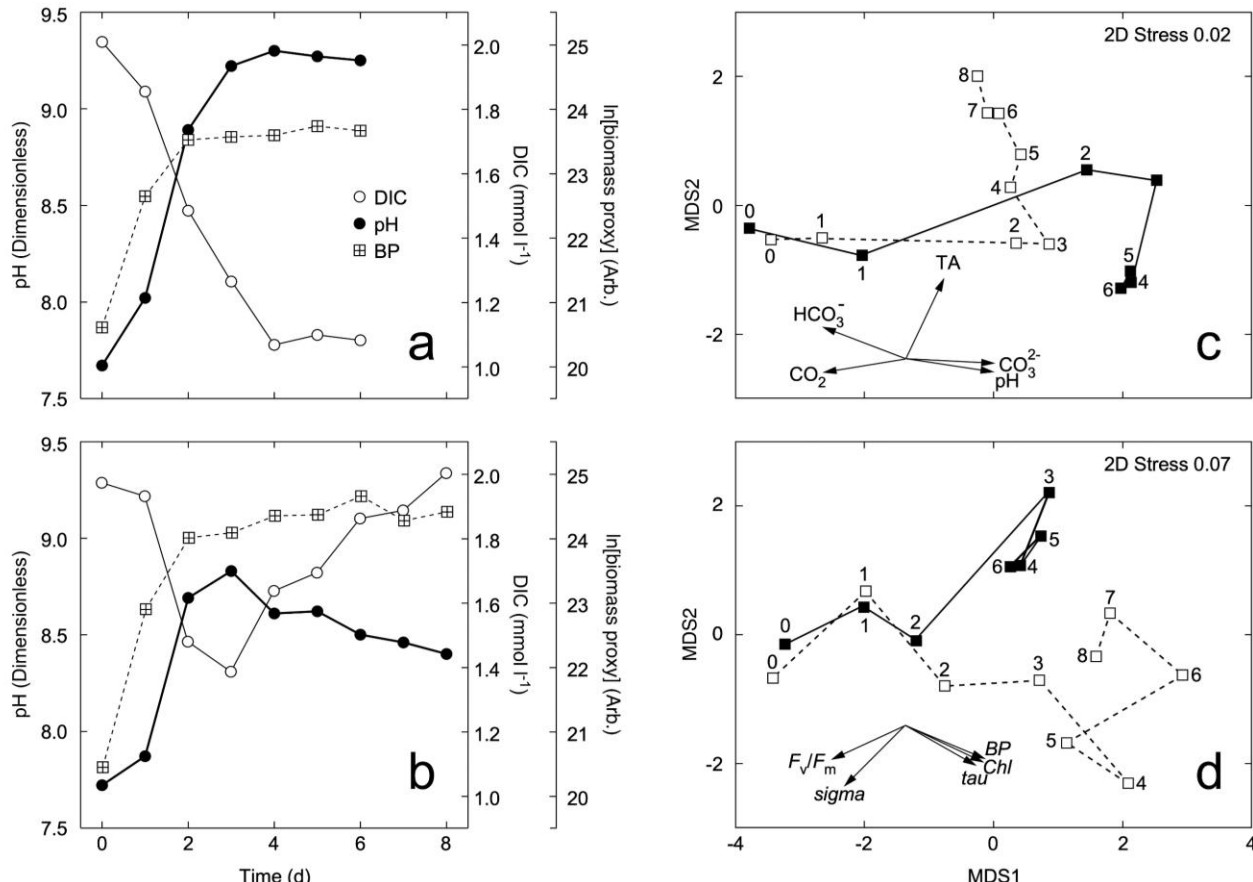

*Figure 3*: Changes in pH, DIC, and a proxy for biomass (BP) in cultures of the diatom *Thalassiosira pseudonana* that

were sealed (a) or aerated (b) during growth (see Supplement 2 for details). Metric multidimensional scaling plots are of

(c) carbonate system variables and (d) biotic variables describing phytoplankton abundance and physiological status. Data

were normalized by variable prior to analysis. The resemblance measure between samples is Euclidian distance. Closed

symbols are for the sealed culture; open symbols are for the aerated culture. Numbers by the symbols refer to time (d)

from inoculation. Vectors are for the factor loadings on input variables in the ordination. Note the divergence of trajectories

in both the abiotic and biotic variables following exponential-phase growth (Day 0-1).

There is clear separation in the mMDS (Figure 3) between the aerated and pH-drift cultures following the period of exponential growth (Days 0-1) with respect to both the carbonate system parameters (Figure 3c) and biological parameters describing abundance and physiological status (3d). These results demonstrate that for the test organism, the biological responses to alkalization depend in large part on $CO_2$ in-gassing, so the response to mineral-based OAE and subsequent in-

 gassing could not necessarily be inferred from pH drift experiments.

### 3.3 Assessing the effects of short- and long-term alkalization on viability, growth, and photosynthetic competence in two coastal phytoplankton

#### 3.3.1 Response to chronic elevated alkalinity in *Thalassiosira pseudonana* and *Diacronema lutheri*

The average value of $F_v/F_m$ at mid-exponential phase for each initial pH tested is shown in Figure 4a. Mid-exponential phase
 was chosen because the cultures had been exposed to the elevated pH for approximately 2 days but were not yet experiencing nutrient limitation during stationary phase that would impact $F_v/F_m$ (Kolber et al., 1998). The trends are not significant (p>0.05), based on linear regression.

Variations in $\mu_m$ (Equation 5) in each treatment are illustrated in Figure 4b. The cultures were exposed to the elevated pH for
 a period of 8 days in batch culture. The pH-dependence was calculated using Equations 2a & c with $\mu_m$ (rather than $\mu$) as the dependent variable. In both cases, the fits were significant (p<0.05). The threshold values of pH above which the exponential growth rates declined were $8.59 \pm 0.06$ for *T. pseudonana* and $8.68 \pm 0.20$ for *D. lutheri*. These thresholds align with the thresholds calculated from the literature in Section 2.3.

#### 3.3.2 Response to transient elevated alkalinity

 The effects of transient alkalization on the proportion of functional PSII reaction centres, $F_v/F_m$, and the maximum specific growth rate, $\mu_m$, were examined with the same methods as the chronic exposure. The average value $F_v/F_m$ for *D. lutheri* was 0.43, indistinguishable from the values in the untreated parent culture, and there was no significant trend (p>0.05) with the transient elevation in pH (Figure 4c). There was a significant trend for *T. pseudonana* (Figure 4c). The data were fit with the biphasic model (Equation 2) and the threshold pH at which there was a reduction was estimated as $8.86 \pm 0.24$.

The $10^{-3}$ dilution from the SDC – MPN assays was used to calculate $\mu_m$, the maximum (exponential phase) growth rate, for each treatment because it was the lower dilution common to all treatments. Note that at a $10^{-3}$ dilution, even the highest hydroxide addition would have been diluted back to background concentrations, so these samples were growing in the original seawater medium. There was no discernible impact of transient exposure to high alkalinity on $\mu_m$ in either species (Figure 4d).
 Neither a linear regression nor the bilinear model (Equation 2) gave statistically significant fits to the data (p>0.05).

The effect of 10-minute exposure to elevated alkalinity and pH on viability in *T. pseudonana* and *D. lutheri* are illustrated in Figure 5. In both cases, there is no evidence for an effect of transient exposure to high alkalinity on viability: Type 1 regressions of RV on pH were not significant ($p > 0.05$). Even with high replication and multiple tiers of dilution, the 95% confidence intervals span about an order of magnitude. In all samples, RV of 1 — the value at which the concentration of viable cells is equal to the total cell concentration — is within the 95% CI of the estimate.

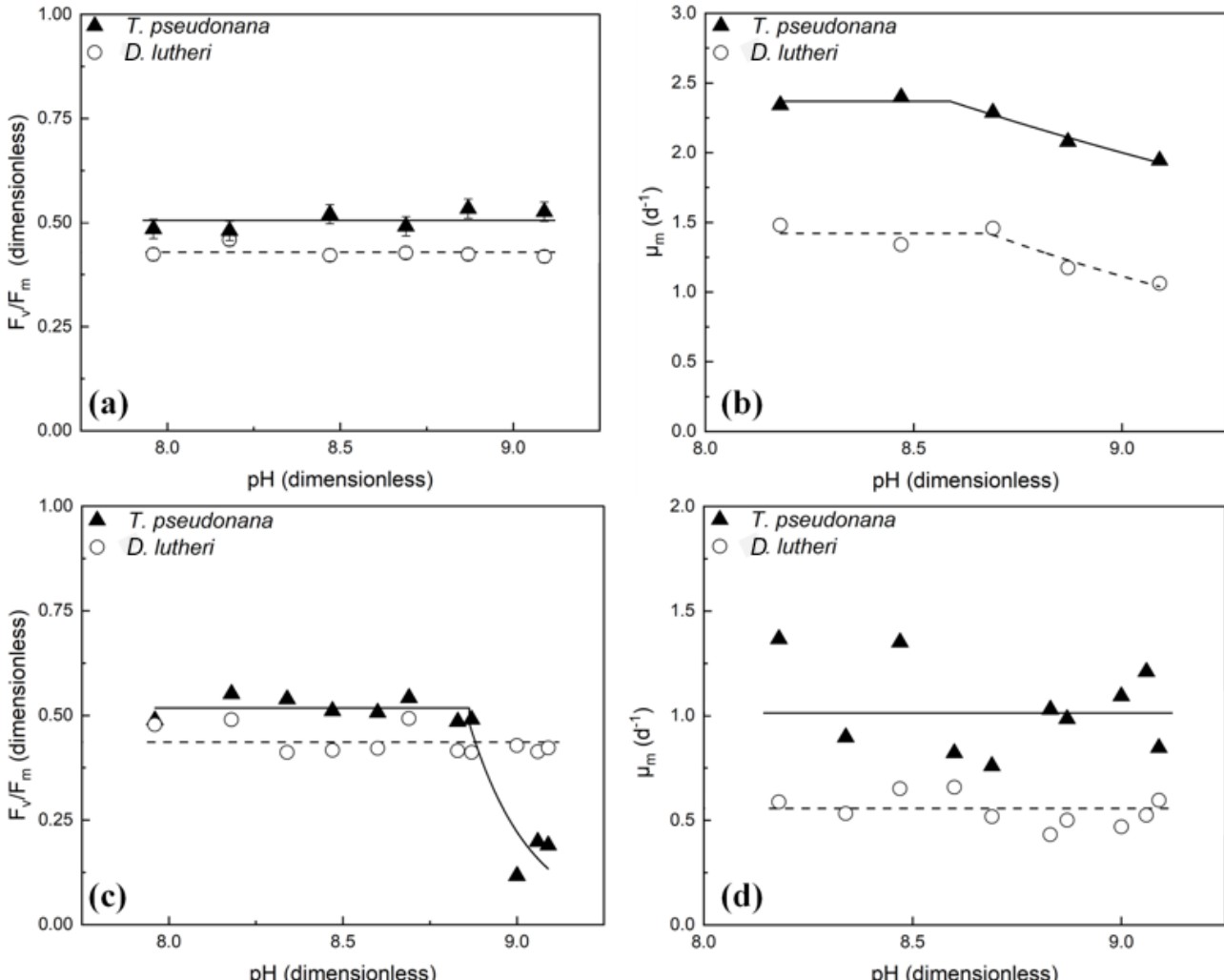

*Figure 4:* Variation in (a) $F_v/F_m$ in mid-exponential phase and (b) $\mu_m$ during chronic exposure to elevated pH in *T. pseudonana* and *D. lutheri*. There was no significant trend in $F_v/F_m$ for either species (p>0.05). The fits to Equations 2 are shown in (b). Measurements of (c) the quantum yield of PSII electron transport, $F_v/F_m$, a measure of the proportion of functional reaction centres, and (d) the maximum fluorescence-based specific growth rate, $\mu_m$, following 10-minute exposure to elevated alkalinity and pH in *T. pseudonana* and *D. lutheri*. Fluorescence was measured 1-2 hours following exposure. Estimates of maximum growth rates are based on fits of the growth curve in samples that were diluted to $10^{-3}$ in the SDC–MPN assay. There was no significant trend in the data for $F_v/F_m$ in *D. lutheri* nor for $\mu_m$ in either species. The dashed lines are the mean values. The reduction in $F_v/F_m$ in *T. pseudonana* was fit to Equation 2 (dashed line). The estimated threshold pH for reduced $F_v/F_m$ is $8.86 \pm 0.24$.


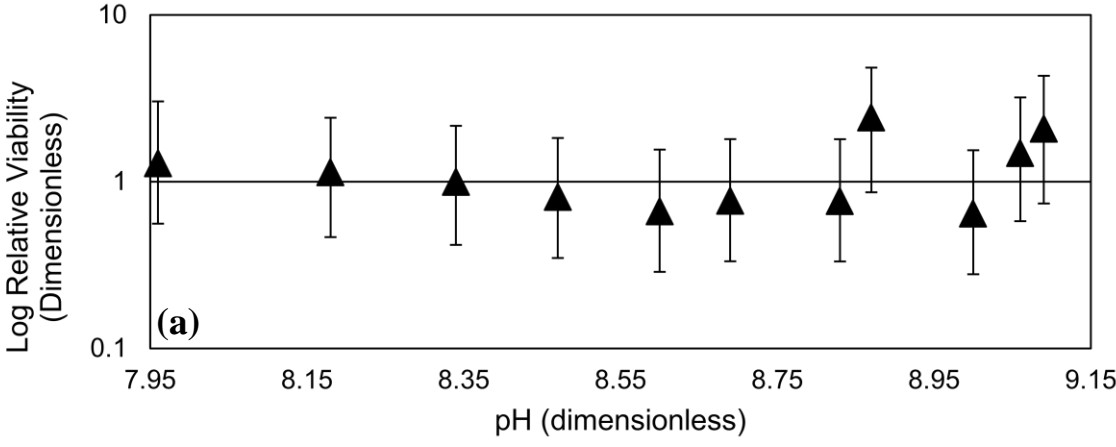

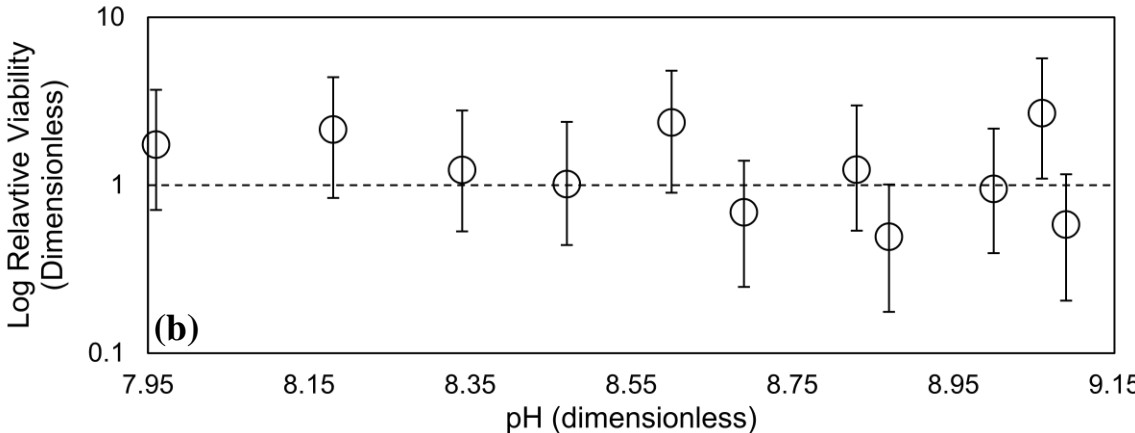

*Figure 5:* Dose-response curves showing the effect of transient exposure to increasing alkalinity pH on relative viability, RV, of a) *T. pseudonana* and b) *D. lutheri*. Error bars are the 95% CI, and the dashed line is a RV of 1 (*i.e.,* no change). Regressions of RV on pH were not significant (p>0.05).

## 4 Discussion and Conclusions

Where OAE is based on unequilibrated alkalization, it will result in a rise in both alkalinity and pH. We have tested potential responses through a combination of data analysis and experimental manipulations, under conditions in which the pH increase is accompanied by DIC drawdown and in conditions in which the two are largely decoupled. The synthesis of studies from the
literature suggests that prolonged long-term increases in pH to 8.2–8.3 would not affect the growth rates of the majority of species studied. This result is not surprising considering the natural variations of pH in a marine system tend to fluctuate much more than this (Oberlander, 2023). The mechanism and potential impacts of OAE allow for $CO_2$ invasion to restore the

equilibrium concentration of $CO_2$ after the conversion of existing $CO_2$ into bicarbonate. Where the approach is not calibrated to raise alkalinity without perturbing pH, OAE will result is a rise in pH with adjustment of the DIC speciation towards

bicarbonate and carbonate, followed by in-gassing of $CO_2$ to restore the equilibrium. If in-gassing is restricted, the rise in pH will be accompanied by a drawdown in DIC if there is an excess of photosynthesis versus respiration. Comparisons of alkalized cultures with high DIC (semi-continuous cultures or aerated batch cultures) and those in which there was a drawdown (sealed pH-drift cultures) show that in some cases there is no difference in the resulting growth rates (the dinoflagellates *Ceratium furca* and *C. fusus*; Figure 2) but in others it is pronounced (the congeneric dinoflagellate *C. tripos* and the diatom *Thalassiosira*

*pseudonana*; Figures 2 and 3). The potential effects of OAE are therefore likely to depend on the degree to which changes in pH and DIC are decoupled. This suggests that evaluation of unequilibrated OAE in the context of pH-drift cultures should be completed with careful consideration of the associated changes in the DIC availability.

The analysis of prior studies of pH-dependant growth rates indicates that dinoflagellates are statistical outliers among the

taxonomic groups and may be more sensitive to alkalization than most others (Table 2; Supplement 1). This might be because there is more complete sampling of this group than of the others. They were the primary focus for these researchers' studies (Hansen, 2002; Hansen et al., 2007; Berge et al., 2010; Berge et al., 2012; Søderberg & Hansen, 2007), which were prompted by the observation that dinoflagellates dominated the summer assemblages in nearby Mariager Fjord when pH could reach up to a value of 9 between the months of May and August. Their higher sensitivity to raised pH, when compared with the other

species investigated in Table 2 and Figure 1, might also reflect fundamental differences in the dinoflagellates' physiology and ecology. Rost et al. (2006) investigated 3 species of dinoflagellates, some tested in the pH-drift studies, demonstrated that they had robust CCMs that were dominated by bicarbonate rather than $CO_2$ transport, even at high pH (8.5–9.1), and concluded that the CCM was unlikely to be limiting to growth. This suggests that their higher sensitivity to alkalization is not related to photosynthetic carbon uptake, in spite of the low $CO_2$ affinity of the L2 isoform of Rubisco (Iñiguez et al., 2020) found in

peridinin-containing dinoflagellates. Their dominance at times when their sensitivity to pH suggests they should be the most impacted might be attributed to their frequent use of mixotrophy (reviewed by Stoecker et al., 2017), the combination of photosynthesis and feeding rather than photosynthesis alone, a trait that confers an advantage for nutrient acquisition during stratification events (Margalef, 1978). There was no indication that the dinoflagellate cultures were fed during the pH-drift experiments. If so, the increased sensitivity might not affect growth in natural environments where feeding is possible. In short,

the apparent sensitivity of dinoflagellates to elevated pH should not be extrapolated to natural assemblages subjected to OAE without further study of its effect on feeding and mixotrophic growth.

A last reason for caution in extrapolating the pH-drift responses lies with the most probable scenario for conducting unequilibrated OAE in coastal waters. In this scenario, discharge in the nearshore will likely be into strong lateral flow. Under

these conditions, exposure to elevated pH would be relatively short until such time as cumulative discharge of alkalinity raised the pH of the entire water body. The combination of dilution by transport and reaction of the alkalinity with $CO_2$ (followed by

$CO_2$ invasion) would greatly reduce the timescale of exposure to elevated pH. With this in mind, we conducted experiments to compare the response of *Thalassiosira pseudonana* and *Diacronema lutheri* to chronic (longer than 5 hours) and transient (10-minutes) exposure to elevated pH and alkalinity.


The threshold values for reductions in growth rate with chronic exposure (Figure 4b) were 8.59±0.059 and 8.68±0.199 for *T. pseudonana* and *D. lutheri*, respectively. These align with the average threshold values observed in Figure 1d for diatoms (8.23 – 9.56) and prymnesiophytes (8.47 – 8.69) measured in pH-drift experiments, as expected from the fact that they were performed under comparable conditions (sealed cultures with minimal headspace for gas exchange). Although growth rates

were reduced, there was no evidence of a reduction in photosynthetic efficiency in either species, as inferred from $F_v/F_m$ measured in the midpoint of exponential phase growth, indicating that the growth-limiting step was downstream of light harvesting and charge separation in PSII. Gately et al. (2023) also measured $F_v/F_m$ values for treated cultures within the range expected for nutrient-replete cells of *Emiliania huxleyi* and *Chaetoceros sp.*, however their growth rates were not statistically different from the controls, as they were in this study for both species (Figure 4b). This is likely due to a lower "high" pH (8.42

– 8.44) compared with this study (9.09).

Transient exposure to elevated pH did not have a statistically significant impact on $F_v/F_m$ for *D. lutheri*, however there was a significant reduction in *T. pseudonana* at the highest pH values tested (8.87 – 9.09; Figure 4c). The lack of an effect under chronic exposure to elevated pH in the same species suggests that it takes between 2 hours and several days for recovery

(photorepair) of *T. pseudonana* acclimated to elevated pH. There was no evidence for persistent reductions in growth rate, nor viability, measured in the days following transient exposure. We note that at the pH in our experiments, the dominant carbon species would be carbonate and would be at a greater risk of precipitating into calcite, thus crossing the threshold for the saturation states for calcite ($\Omega_{calc}$) and/or aragonite ($\Omega_{arag}$; Schulz et al., 2023). This could lead to reductions in the availability of bicarbonate for the CCM and might account for the reductions in growth rates observed in Figure 4. Elevating the pH this

high is highly undesirable from the perspective of OAE, as calcite precipitation releases $CO_2$ rather than traps it (Schluz et al., 2023; Moras et al., 2022; Hartmann et al., 2023). It would be possible through the accidental discharge of a concentrated hydroxide slurry, though, so the degree of impairment represents a worst-case scenario rather than the response under expected discharge conditions.

The transient exposures were conducted with only two species, so cannot be considered representative of phytoplankton in general. However, conducting experiments with natural assemblages in micro/mesocosms, such as the studies by Guo et al. (2024), Ferderer et al. (2022), and Paul et al. (2024), can tell us more about the complex community interactions that may be impacted. Both Guo et al. (2024) and Ferderer et al. (2022), who similarly to this study used an unequilibrated OAE approach, observed a lack of a bloom delay in their microcosm experiments, which was contrary to their initial predictions. However,

Paul et al. (2024), who used an equilibrated OAE approach, note that there were only substantial blooms in certain mesocosms

and that the reason for this is not clear in their experiment. Additionally, Guo et al. (2024) state that they observed a shift in the community composition of olivine and slag treated microcosms, with microphytoplankton becoming the dominant group, while Ferderer et al. (2022) found that alkalinity from NaOH did have a significant impact on the phytoplankton groups present. Overall, there is still much work to be done on investigating the impacts of OAE at the individual species level to complex

community interactions.

This study clearly shows that there are significant differences between the effects of transient and long-term exposure to elevated pH. There are significant changes in growth rates with chronic exposure to elevated pH that are consistent with reports from the literature but are not observed following transient exposure to the same range of high pH. If consistent results are

found across taxa or in mixed assemblages, the estimates of threshold pH at which growth rates were reduced — which were all based on chronic exposure in pH-drift or semi-continuous culture — are likely to overestimate the potential impact of OAE if the alkalinity is diluted on timescales of 10 minutes to 1-2 hours following discharge into the receiving waters.

*Data Availability*

Data will be made available upon request.

*Author Contributions*

JLO and HLM designed the experiments and JLO, MB, and CL conducted them. HLM supervised the study. JLO was responsible for the literature review, data digitization, data analysis, and statistical analysis. JLO and HLM prepared the

manuscript.

*Competing Interests*

The contact author has declared that none of the authors have any competing interests.

*Acknowledgements*

This research was funded through the ClimateWorks Foundation and the Thistledown Foundation donations, and the Natural Sciences and Engineering Research Council of Canada grant CRDPJ 520352-17. The authors would like to thank Mikeala Ermanovics and Rose Latimer for assistance with conducting experiments and culture maintenance.

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
