# Peer review of "Assessing the impacts of simulated Ocean Alkalinity Enhancement on viability and growth of near-shore species of phytoplankton"

_EGUsphere, 2024_

## Author Comment (AC1)

**RC1: Anonymous Referee #1**

We thank the reviewers for their constructive detailed, and very helpful analyses of the draft manuscript. Responses to their comments are given below. The majority of comments can be grouped into 4 themes:

- 1. Inadequate description of methods and their disjointed presentation in different parts of the manuscript;
- 2. Inadequate referencing of the (rapidly growing) literature on OAE;
- 3. A lack of clarity in considering pH vs DIC effects, particularly in the analysis of prior studies using pH-drift experiments; and
- 4. Overly narrow considerations of potential approaches (coastal vs open ocean; equilibrated vs non-equilibrated) to OAE.

We are confident that we can address the reviewers' concerns and, with their guidance, can revise and improve the manuscript. Detailed responses to their comments are given below.

**General Comments:**

There are three main conclusions made within this study. The first is that there were no significant impacts on the viability or growth rate of the two species assessed within the short term exposure to elevated alkalinity levels. This is to be expected as levels of OAE that would likely result in an impact within 10 minutes of exposure would be significantly higher than what is logistically possible or safe in terms of secondary precipitation. The second is that longer term exposure to elevated alkalinity resulted in a decrease in growth rate. This is an interesting finding as many other studies thus far have found little evidence to suggest that there will be significant impacts to the growth of phytoplankton as a result of OAE. The third is that within the current literature evidence suggests that approximately 50% of species could be significantly impacted by pH increases in line with those expected as a result of OAE. This provides an excellent summary of the current knowledge of pH impacts however it is unclear as to whether these impacts are a result of pH (H+ concentrations) or CO2 concentrations, of which the latter is expected to be the most important factor in regard to the ecological impacts of OAE on phytoplankton.

**Major Comments:**

The manuscript does an excellent job in referencing and discussing the influence of elevated pH on phytoplankton growth and viability. However, citations and discussion around OAE specifically are currently lacking. Further I found it difficult to disentangle exactly how the experiments were run within this manuscript and strongly recommend that the methods/results section be revised so that readers may easily understand what the authors have accomplished here.

We agree that referencing other studies on OAE needs to be updated and improved throughout the manuscript and that the structure of several of the sections (specifically sections 3 and 4) should be revised for clarity and accessibility. We provide more detail below.

**Specific Comments:**

 Lines 8 – 9: There are many methods for achieving OAE and only mineral based methods could be considered to "mimic" the natural weathering of alkali minerals. Furthermore, enhancing alkalinity does not lead to the sequestering of atmospheric CO2 but decreases oceanic CO2 concentrations allowing for atmospheric CO2 to be sequestered under the correct conditions. I appreciate that this is within the abstract and concise wording is necessary but would advise the authors to consider rewording this sentence.

**Suggested rewording:**

"One proposed NET is Ocean Alkalinity Enhancement (OAE), in which artificially raising the alkalinity favours formation of bicarbonate from  $CO_2$ , leading to a decrease in the partial pressure of  $CO_2$  in the water, and a subsequent invasion of atmospheric resulting in net sequestration of atmospheric carbon."

2. Line 11-12: This sentence is difficult to understand, consider rewording.

**Suggested rewording:**

"The potential impacts of OAE were assessed through an analysis of prior studies investigating the effects of elevated pH on phytoplankton growth rates in pH-drift experiments and by experimentally assessing the potential impact of short-term elevation of pH on the viability and subsequent growth rates of two representative near-shore species of phytoplankton."

3. Line 16 – please provide the actual number of days or range of days.

**Suggested rewording:**

"However, there was a significant decrease in growth rates with long-term (8 days) exposure to elevated pH"

4. Lines 18-19: Within this manuscript the authors have only looked at two species from two different taxonomic groups. This statement is too broad for the results of the experiment conducted here.

We suggest deleting the final sentence of the abstract.

5. Line 31: I caution the authors against the use of the word "natural". All methods of CDR are anthropogenically motivated and furthermore depending on the method of OAE used it may be more or less similar to the natural weathering of alkali minerals (e.g. mineral based OAE vs electrochemical based OAE). Furthermore, it is highly unlikely that OAE would result in these changes as any increase in pH would ideally be negated by an influx of CO2, resulting in an increase in DIC beyond current and pre-industrial concentrations.

**Suggested rewording:**

"Ocean Alkalinity Enhancement (OAE) is one promising NET that involves anthropogenically raising the alkalinity of a parcel of water causing the partial pressure of the  $CO_2$  in that water to decrease. This change leads to either in-gassing of  $CO_2$  from the atmosphere or a reduction in outgassing of  $CO_2$  from the ocean, depending on the initial air-sea gradient. Both scenarios result in a theoretical net reduction of atmospheric  $CO_2$  through storage in the form of bicarbonate  $(HCO_3^2)$  and carbonate  $(CO_3^{2-})$  ions in the ocean (Oschlies et al., 2023)."

6. Line 33: Ideally this would be the case however additions of alkalinity and subsequent in gassing of atmospheric CO2 will lead to increases of both carbonate and bicarbonate. This will depend on

other variables, but without CO2 equilibration concentrations of carbonate would increase significantly more than bicarbonate.

**Suggested rewording:**

"The additional carbon would be stored in the form bicarbonate (HCO3-) and carbonate (CO32-) ions, with the former (which has a residence time of c. 1,000 years in the ocean) favoured after in-gassing of CO2."

 Line 34: "...would likely be...". Also, it is not yet clear which method of OAE will be implemented at large scales. It is more likely that multiple methods will be implemented in different regions e.g. electrochemical in coastal regions and mineral based in pelagic regions.

**Suggested rewording:**

"There are currently several different methods of OAE in development, including mineral- and electrochemical-based methods, with deployment from vessels, through preexisting outfalls, or from placement on beaches. The focus of this study is the mineral-based approach from preexisting outfalls, implementation of which is likely to occur through addition of unequilibrated hydroxide minerals (OH-) to the coastal surface ocean."

8. Line 36: Again, enhancing alkalinity does not necessarily result in bicarbonate formation. For a parcel of water with TA 2100 μmol/kg, DIC 2000 μmol/kg, T 15°C and salinity 35 an increase in alkalinity of 500 μmol/kg would result in changes of bicarbonate – 316.93 μmol/kg, carbonate + 340 μmol/kg. Also, such changes in carbonate chemistry do not necessarily lead to CO2 drawdown this is dependent on several other factors.

**We suggest deleting this sentence.**

9. Lines 84 – 85: The authors discuss changes in pH throughout this section but was there additional criteria to ensure that manipulations of pH resulted in changes in DIC that would be expected of OAE (particularly for those articles not by "Hansen and colleagues")? For example, using varied additions of sodium carbonate and HCl one can achieve similar pH values but drastically different concentrations of TA and DIC. Given this and the expected ranges of pH in OAE one would expect CO2 and not pH to be the driving factor to influence phytoplankton.

We agree that more clarity on the roles of pH vs DIC is critical for interpreting these data. However, most of the papers we included in the analysis did not report both pH and DIC and we did not use DIC as a criterion for including the study in the analysis. Our intention was to illustrate this with the single experiment presented in Figure 3, which contrasts the changes in pH and DIC in pH-drift vs aerated cultures (i.e., without/with in-gassing) and in which the biological responses were most consistently correlated with CO2 in a multivariate analysis. Even so, we argue that mineral alkalization of an outfall (our focus) is likely to have an effect on pH without a significant reduction in DIC and that pH has the potential to be an important determinant of phytoplankton metabolism and growth.

Suggested modification: "...batch cultures without ventilation to replenish  $CO_2$ . It is critical to recognize that in pH-drift cultures such as these, both pH and DIC vary and that the biological response may be due to changes in  $CO_2$  availability as much as, or more than, by pH (see Section 3, below)."

10. Figure 1: The y axis of figure 1a/b are very hard to discern as they overlap and it is unclear which axis label is referring to which set of units on the axis.

Agreed, the figure can be modified (and saved at a higher resolution) for clarity.

11. Figure 1d: It is very difficult to see the median value for the 90%  $\mu$  reduction. Consider changing the colour of this line and/or adjusting the alpha values in this plot.

Agreed.

12. Lines 149 – 148: Is this anticipated maximum pH value for OAE based on changes in pH at the point of alkalinity addition or after dilution within a region?

This pH is the anticipated maximum at the point of addition, and suggest this modification for clarity.

"For all of the species investigated (Figure 1), the median threshold pH is above 8.5, suggesting about 50% of the species would not be impacted by the anticipated maximum pH increase at the point of addition anticipated for a mineral-based addition of OAE from a land-based point source."

13. Lines 174: I recommend the authors adjust the wording of this sentence as OAE does not change DIC concentrations initially so there can be no "resupply" of DIC. More accurately this would be an influx of CO2 increasing DIC beyond what it was prior to OAE.

**Suggested rewording:**

"Most of the studies analyzed in Section 1 did not permit  $CO_2$  in-gassing, as they were conducted using a closed-bottle, batch-culturing method. However, for OAE to successfully function as a NET, in-gassing of  $CO_2$  is required to increase the DIC pool."

14. Lines 190 -192: If these observations are made on the basis of the Metric multidimensional scaling plots, please state this.

**Suggested rewording:**

"There is clear separation in the mMDS (Figure 3) between the aerated and pH-drift cultures following the period of exponential growth (Days 0-1), both in the carbonate system parameters (Figure 3c) and in the biological parameters describing abundance and physiological status (3d)".

15. Lines 195 – 196: This is a strong statement to be made from a relatively small comparison. Furthermore, there are various approaches to OAE that would not necessarily result in CO2 influxes similar to those seen here e.g. in areas where CO2 outgassing occurs, or where OAE is added in an equilibrated form. Could the authors provide some references to support this statement?

**Suggested rewording:**

"These results demonstrate that the test organism, the biological responses to alkalization depend in large part on CO2 in-gassing, so the response to mineral-based OAE and subsequent in-gassing could not necessarily be inferred from pH drift experiments." 16. Lines 209 – 213: Please make it clear to the reader at the start of this section whether the cultures were grown in filtered seawater with the addition of nutrients and trace metals? Also, it would be beneficial to disclose the exact location of water collection i.e. xxx kms offshore via boat or pump. In addition, it is stated that "...fresh media in mid-exponential phase..." is this for the experimental cultures or the maintenance of culture pre experiment?

**Suggested rewording:**

"The cultures were maintained in 40-mL volumes of sterile-filtered f/2 (Guillard, 1975) or L1 (Guillard & Hargraves, 1993) seawater medium, and diluted into fresh media in mid-exponential phase in a laminar flow hood. The seawater was collected by pump about 100 m offshore from the National Research Council of Canada's Marine Institute at Ketch Harbour, NS, and tangential flow filtered on collection. It was refiltered through a 0.2- $\mu$ m capsule filter (Cytiva Whatman Polycap Disposable Capsules: 75TC) and nutrient-enriched in autoclaved glassware or in sterile cell culture plates. Prior to experimentation, parent cultures were fully acclimated to the experimental growth conditions, continuous illumination at c. 190  $\mu$ mol photons m-2 s-1 at a temperature of 18 ± 1°C, by maintaining them in balanced growth in semi-continuous culture (MacIntyre & Cullen, 2005)"

Comments 18 and 29 make it clear that a major reorganization of the Methods would be helpful to the reader. We propose to address this by consolidating all of the experimental methods into a new, separate section, with the differences in set-up between the different experiments tabulated for ease of comparison. The table would include the experimental ID, treatment (alkalization as concentration and duration of exposure), culture volume, presence/absence of aeration, measurement parameters and frequency. The revised Methods would also include a better explanation of the rationale for picking the studies compared in Section 2: a literature survey in which our search terms included "phytoplankton AND (alkalinization OR "high pH")". Studies found in the search were gated to include those in which cultures were maintained under pH drift conditions, with culture medium ensuring that the media would have DIC:DIN below the Redfield Ratio, and with time-series (growth curves) of phytoplankton abundance and pH.

17. Line 250: The authors previously used a dark acclimation of 30 minutes, is there a reason for the change to 20 minutes?

This was a typographical error. The dark acclimation time was 30 minutes.

18. It is not clear how the transient exposure cultures discussed in section 4.3.2 are setup. Are these setup the same as those for the chronic exposure experiment? The authors also mention cultures measured after 1-2 hours, are these the same cultures or separate to the chronic and transient exposure cultures? This confusion may come from the layout in which the methods/results are presented. I am hesitant to suggest changes to this but it would be beneficial to the reader if the authors could be overly obvious in the explanation of how cultures were setup and whether cultures measured after 1-2 hours, several days and/or 10 minutes are the same cultures or separate cultures.

Please see our response to Comment 16.

19. Lines 294 - 296: The authors state "...there was no significant trend with the transient elevation in pH transient elevation in pH (Figure 4c)." However, the figure caption for figure 4 states "...measured after exposure to elevated alkalinity and pH for 1-2 hours...". Is the transient elevated alkalinity assessing the effect after 10 minutes as stated in line 286 or 1-2 hours?

We agree that the presentation is confusing. The cultures were exposed to the elevated alkalinity for 10 minutes before being diluted with the SDC-MPN method into untreated media. However, due to the time needed to finish the dilution series, and the constraint on personnel, the subsample of culture used to measure  $F_{\nu}/F_m$  could not be read immediately following the 10 minute exposure and thus were read between 1-2 hours following exposure.

Suggested rewording in the legend to Figure 4:

"Measurements of (c) the quantum yield of PSII electron transport,  $F_{\nu}/F_m$ , a measure of the proportion of functional reaction centres, and (d) the maximum fluorescence-based specific growth rate, µm, following 10-minute exposure to elevated alkalinity and pH in *T. pseudonana* and *D. lutheri*. Fluorescence was measured 1-2 hours following exposure. Estimates of maximum growth rates are based on fits of the growth curve in samples that were diluted to10-3 in the SDC– MPN assay. There was no significant trend in the data for  $F_{\nu}/F_m$  in *D. lutheri* nor for µm in either species. The dashed lines are the mean values. The reduction in  $F_{\nu}/F_m$  is 8.86 ± 0.24."

20. Figures should be introduced in the order they appear in the text e.g. figure 5 is introduced in line 287 while figure 4c is introduced in line 296.

We thank the reviewer for pointing out this placement error and will switch the order of Figures 4 and 5. We believe that keeping all subfigures in figure 4 is beneficial for the reader to make visual comparisons in the data.

21. Lines 310-311: This sentence needs to be revised in line with comment 8.

Suggested rewording:

"The mechanism of unequilibrated OAE in regions of the ocean where the initial air-sea gradient favours in-gassing of  $CO_2$ , allows for the invasion of  $CO_2$  to occur and re-equilibrate acress the air-sea interface. The additional  $CO_2$  is then stored in the form of bicarbonate and carbonate (Oschlies et al., 2023)."

22. Lines 311 - 314: It would be beneficial if the authors could provide some reasoning behind why there are differences between these methods and how to abate these differences.

The differences presumably reflect a difference in the physiology of the different species, but notably occur within a single genus, so don't fit on high-level taxonomic differences (e.g., diatoms vs dinoflagellates). We have no data from which to infer a mechanistic basis and prefer not to make completely unsupported suggestions of cause.

23. Lines 354 – 356: This sentence is confusing particularly "...the dominant carbon species would be carbonate and calcite would begin precipitating...". Do the authors mean that carbonate is the dominant form of carbon and would precipitate into calcite? Traditionally when discussing precipitation omega values are provided, this would be beneficial here as many other articles discuss calcite precipitation in this form see Moras et al. (2022) and Schulz et al. (2023).

The intention of this sentence was to state that carbonate is the dominant form of carbon and would precipitate into calcite at this pH. We agree that the wording here is slightly confusing and have reworded to get our point across more clearly, as well as including reference to crossing the omega thresholds. We believe that specific omega values are outside the scope of this work. Suggested rewording:

"We note that at the pH range in our experiments, the dominant carbon species would be carbonate and would be at a greater risk of precipitating into calcite, thus crossing the threshold for the saturation states for calcite ( $\Omega_{calc}$ ) and/or aragonite ( $\Omega_{arag}$ ; Schulz et al., 2023). This could lead to reductions in the availability of bicarbonate for the CCM and might account for the reductions in growth rates observed in Figure 4."

24. Lines 356 – 357: It would be beneficial to cite some of the many article's discussing the efficiency of OAE and impacts of calcite precipitation here.

Agreed. We will refer to Schulz et al. (2023), Moras et al. (2022), and Hartmann et al. (2023).

25. There have been numerous studies assessing the impact of OAE on natural assemblages of phytoplankton and cultures in recent years e.g. Gately et al. (2023) and Guo et al. (2023). Although the authors discuss changes in pH there is no significant discussion surrounding the numerous papers on the impact of OAE on phytoplankton. The manuscript is significantly lacking in this regard and inclusion and discussion of such articles within the introduction and/or discussion would help to improve this manuscript.

Agreed. We will refer to: Gately et al. (2023), Iglesias-Rodríguez et al. (2023), Guo et al. (2023), Paul et al. (2024), Hutchins et al. (2023), Ferderer et al. (2022), and Subhas et al. (2022).

26. The authors discuss initial pH values, were there also end measurements for pH or prior to media refreshment? Volumes used were relatively low and cultures were grown into stationary phase, as such a significant change in pH over the duration of the experiment would be expected. I understand that during the SDC – MPN assay it would be expected that in gassing occurred maintaining pH at a relatively stable level, however this is not clear for the other experiments.

We did not collect measurements for pH at the end of the experiments conducted in tubes because, as noted, there would be a significant change in pH over the experiment caused by the phytoplankton themselves. These changes would be more gradual as a consequence of the phytoplankton growth.

27. Line 241-242: Was alkalinity measured or was it calculated? If it was calculated was this based on the additions of NaOH or another measured carbonate chemistry parameter? It would be beneficial to the manuscript to add measured or alkalinity values calculated from a second carbonate chemistry parameter, as many articles discuss OAE in terms of alkalinity increases (μmol/kg) and not pH increases. The alkalinity values reported are calculated based on the measured pH and known DIC values. Suggested rewording (line 241-242):

"These additions increased the initial concentration of total alkalinity, 2168  $\mu$ mol L-1, by 0-1084  $\mu$ mol L-1. (All reported values of alkalinity are calculated from measured values of pH and DIC.) The culture and NaOH..."

28. The quality of Figure S1 is extremely poor and makes it difficult to read. In addition, Figures S1 c,d appear to be concentrations of carbon species (CO2, HCO-3 and CO32-) not DIC as stated in the figure caption.

We apologize for the quality of the figure, which was degraded in successive file conversions. It can be restored to original resolution and the caption corrected to accurately distinction between the carbon species and DIC.

29. It was difficult to distinguish between the experiments, variables measured, and methods used here. I strongly recommend the authors increase the clarity of the text discussing the experimental methods. For example, lines 209 – 210 "The cultures were grown in 40-mL volumes of f/2 (Guillard, 1975) or L1 (Guillard & Hargraves, 1993) seawater medium, and diluted into fresh media in mid-exponential phase in a laminar flow hood." Was this done for all experimental culture's transient, chronic and those in the SDC-MPN assay? Or was this only prior to the experiment during the acclimation? Am I correct to understand that there are separate cultures with separate maintenance methods i.e. the SDC-MPN cultures, the chronic and transient? A strict section or table outlining the experiments conducted and their common and/or differences in methodology would be greatly beneficial, as it is currently difficult to tease apart exactly how this experiment was conducted.

We agree that the current structure of Section 4 would benefit from revisions for clarity and thank the reviewer for the suggestions for doing so. We believe that the change in structure recommended in response to Comment 16 will aid in the reader's comprehension of the methods/variables for each experiment and that a summary tabulation of differences would facilitate this.

30. Lines 10-12: The authors introduce the analysis of prior studies here but fail to state any results from this within the abstract. This is a major component of the manuscript and requires a section detailing the results within the abstract.

**Suggested addition to the abstract:**

"The analysis of prior studies indicates wide variability in the growth response to elevated pH within and between taxonomic groups, with about 50% of species expected to not be impacted by pH increase expected unequilibrated mineral-based OAE. To the extent that the growth responses reflect (largely unreported) parallel reductions in DIC availability, the susceptibility may be reduced for OAE in which  $CO_2$  in-gassing is not prevented."

**References:**

Ferderer, A., Chase, Z., Kennedy, F., Schulz, K. G., and Bach, L. T.: Assessing the influence of ocean alkalinity enhancement on a coastal phytoplankton community, Biogeosciences, 19, 5375–5399, https://doi.org/10.5194/bg-19-5375-2022, 2022.

Gately, J. A., Kim, S. M., Jin, B., Brzezinski, M. A., and Iglesias-Rodríguez, M. D.: Coccolithophores and diatoms resilient to ocean alkalinity enhancement: A glimpse of hope?, Science Advances, 9, eadg6066, doi:10.1126/sciadv.adg6066, 2023.

Guo, J. A., Strzepek, R. F., Swadling, K. M., Townsend, A. T., and Bach, L. T.: Influence of ocean alkalinity enhancement with olivine or steel slag on a coastal plankton community in Tasmania, Biogeosciences, 21, 2335–2354, https://doi.org/10.5194/bg-21-2335-2024, 2024.

Hartmann, J., Suitner, N., Lim, C., Schneider, J., Marín-Samper, L., Arístegui, J., Renforth, P., Taucher, J., and Riebesell, U.: Stability of alkalinity in ocean alkalinity enhancement (OAE) approaches – consequences for durability of CO2 storage, Biogeosciences, 20, 781–802, https://doi.org/10.5194/bg-20-781-2023, 2023.

Hutchins, D. A., Fu, F.-X., Yang, S.-C., John, S. G., Romaniello, S. J., Andrews, M. G., and Walworth, N. G.: Responses of globally important phytoplankton species to olivine dissolution products and implications for carbon dioxide removal via ocean alkalinity enhancement, Biogeosciences, 20, 4669–4682, https://doi.org/10.5194/bg-20-4669-2023, 2023.

Iglesias-Rodríguez, M. D., Rickaby, R. E. M., Singh, A., and Gately, J. A.: Laboratory experiments in ocean alkalinity enhancement research, in: Guide to Best Practices in Ocean Alkalinity Enhancement Research, edited by: Oschlies, A., Stevenson, A., Bach, L. T., Fennel, K., Rickaby, R. E. M., Satterfield, T., Webb, R., and Gattuso, J.-P., Copernicus Publications, State Planet, 2-oae2023, 5, https://doi.org/10.5194/sp-2-oae2023-5-2023, 2023.

Moras, C. A., Bach, L. T., Cyronak, T., Joannes-Boyau, R., and Schulz, K. G.: Ocean alkalinity enhancement – avoiding runaway CaCO3 precipitation during quick and hydrated lime dissolution, Biogeosciences, 19, 3537–3557, https://doi.org/10.5194/bg-19-3537-2022, 2022.

Oschlies, A., Bach, L. T., Rickaby, R. E. M., Satterfield, T., Webb, R., and Gattuso, J.-P.: Climate targets, carbon dioxide removal, and the potential role of ocean alkalinity enhancement, in: Guide to Best Practices in Ocean Alkalinity Enhancement Research, edited by: Oschlies, A., Stevenson, A., Bach, L. T., Fennel, K., Rickaby, R. E. M., Satterfield, T., Webb, R., and Gattuso, J.-P., Copernicus Publications, State Planet, 2-0ae2023, 1, doi:10.5194/sp-2-0ae2023-1-2023, 2023.

Paul, A. J., Haunost, M., Goldenberg, S. U., Hartmann, J., Sánchez, N., Schneider, J., Suitner, N., and Riebesell, U.: Ocean alkalinity enhancement in an open ocean ecosystem: Biogeochemical responses and carbon storage durability, EGUsphere [preprint], https://doi.org/10.5194/egusphere-2024-417, 2024.

Schulz, K. G., Bach, L. T., and Dickson, A. G.: Seawater carbonate chemistry considerations for ocean alkalinity enhancement research: theory, measurements, and calculations, in: Guide to Best Practices in Ocean Alkalinity Enhancement Research, edited by: Oschlies, A., Stevenson, A., Bach, L. T., Fennel, K.,

Rickaby, R. E. M., Satterfield, T., Webb, R., and Gattuso, J.-P., Copernicus Publications, State Planet, 2-oae2023, 2, https://doi.org/10.5194/sp-2-oae2023-2-2023, 2023.

Subhas, A. V., Marx, L., Reynolds, S., Flohr, A., Mawji, E. W., Brown, P. J., and Cael, B. B.: Microbial ecosystem responses to alkalinity enhancement in the North Atlantic Subtropical Gyre, Frontiers in Climate, 4, https://doi.org/10.3389/fclim.2022.784997, 2022.

---

## Author Comment (AC2)

*General Comments:*

I read with pleasure the paper: "Assessing the impacts of simulated Ocean Alkalinity Enhancement on viability and growth of near-shore species of phytoplankton" by J. Oberlander and coauthors. This manuscript is of high interest for the assessment of OAE. The author digs into the available data set from the previous study that investigated the impact of increased pH on phytoplankton (outside the context of OAE) and performs culture experiments on two species, Thalassiosira pseudonana and Diacronema lutheri with two different approaches (batch culture and semi-continuous) to trace the (very) short impact of increase pH and the long(er) impact in the context of OAE.
That said, I think the manuscript needs significant revision before it is suitable for publication.

*Major Comments:*

It was hard to follow the structure of the article. Some chapters and subchapters do not follow the standard structure of an article to the point that methodologies, statistics analyses and results are mixed.

We agree that a major reorganization of the Methods would be helpful to the reader. We propose to address this by consolidating all of the experimental methods into a new, separate section, with the differences in set-up between the different experiments tabulated for ease of comparison. The table would include the experimental ID, treatment (alkalization as concentration and duration of exposure), culture volume, presence/absence of aeration, measurement parameters and frequency. The revised Methods would also include a better explanation of the rationale for picking the studies compared in Section 2: a literature survey in which our search terms included "phytoplankton AND (alkalinization OR "high pH")". Studies found in the search were gated to include those in which cultures were maintained under pH drift conditions, with culture medium ensuring that the media would have DIC:DIN below the Redfield Ratio, and with time-series (growth curves) of phytoplankton abundance and pH.
(Reviewer 1, Comment 16)

The analysis of previous studies is almost at the beginning of the article, but it is not discussed adequately in the discussion and conclusion chapters. Some literature is omitted throughout the article. What about other studies made for ocean acidification that tested however higher pH values e.g. Bach et al., 2015? Or some of the Riebesell et al., studies on coccolithophore?

With respect to the analysis of prior studies in the Discussion and Conclusion, we will work to correct this by including a paragraph outlining the conclusions from the analysis as well as discussion further explaining the connection with OAE.
In regard to studies omitted in the analysis, we focused primarily on laboratory studies that included direct relationships between increasing pH and the concentration of cells or growth rate of the phytoplankton. The studies included in this analysis had this data easily available in their papers (*i.e.*, figures or reported values), and due to time constraints contacting authors to request access to data that was not included was not feasible. We agree that the work conducted by Bach, Riebesell, and many others on ocean acidification is valuable to understanding the potential impacts of OAE. We note that Bach et al. (2015) does include pH and growth rate data for *C. pelagicus*, however because the focus of his study was $CO_2$ it did not appear in our literature search for prior studies investigating elevated pH where our search terms included "phytoplankton AND (alkalinization OR "high pH")". We believe that for the purposes of this manuscript, which this analysis was a single section, that focusing on a subset of the literature that is

consistent in terms of experimental approach and reporting gains in the simplicity of the comparison what it loses in breadth.

In the discussions previous articles related to the response of phytoplankton to the perturbation induced by OAE are not mentioned (e.g. Gately et al., 2023) and in general many references are missing throughout the text.

The same point was made by Reviewer 1. We propose to reference Gately et al. (2023), Iglesias-Rodríguez et al. (2023), Guo et al. (2023), Paul et al. (2024), Hutchins et al. (2023), Ferderer et al. (2022), and Subhas et al. (2022) for studies of the impact of OAE on phytoplankton, and Schulz et al. (2023), Moras et al. (2022), and Hartmann et al. (2023) for the efficiency of OAE and the impacts of calcite precipitation

*Specific Comments:*

**Abstract**
The abstract should be strongly revised. On top of some inaccuracies that I will report in the following lines, the reasons why studying the impact of high pH on phytoplankton is lost within the lines. Moreover, the study is based on two specific species that are important components of the phytoplankton community in specific marine contexts (i.e. near-shore, temperate waters). The final statement (lines 18-19) is therefore too general and misleading. The author should consider the findings of this study and not generalise their results to other taxonomic groups. The analysis of the prior study is lacking in the abstract even if it's an important part of this study. It should be mentioned.

The same problems were noted by Reviewer 1 in comments 1-4 and 30, our proposed changes are as follows:
- Rewording lines 8-9: "One proposed NET is Ocean Alkalinity Enhancement (OAE), in which artificially raising the alkalinity favours formation of bicarbonate from $CO_2$, leading to a decrease in the partial pressure of $CO_2$ in the water, and a subsequent invasion of atmospheric resulting in net sequestration of atmospheric carbon."
- Rewording lines 11-12: "The potential impacts of OAE were assessed through an analysis of prior studies investigating the effects of elevated pH on phytoplankton growth rates in pH-drift experiments and by experimentally assessing the potential impact of short-term elevation of pH on the viability and subsequent growth rates of two representative near-shore species of phytoplankton."
  Rewording line 16: "However, there was a significant decrease in growth rates with long-term (8 days) exposure to elevated pH."
- Deleting lines 18-19
- Addition of the following: "The analysis of prior studies indicates wide variability in the growth response to elevated pH within and between taxonomic groups, with about 50% of species expected to not be impacted by pH increase expected unequilibrated mineral-based OAE. To the extent that the growth responses reflect (largely unreported) parallel reductions in DIC availability, the susceptibility may be reduced for OAE in which CO2 in-gassing is not prevented."

1. Lines 7-8: technologies used twice.

Suggested rewording: "In response, new tools are being developed…"

2. Line 9: OAE is not only mimicking but is enhancing/accelerating the process. Moreover, it should be made more explicit why CO2 is ultimately sequestered from the atmosphere. The link is missing.

    Please see the response above to the general abstract comments for the suggested revision to lines 8-9.

3. Lines 10-11: As mentioned before, the aim of the study and the need to understand the impact of increased pH on primary producers should be better expressed.

    Proposed rewording:
    "The aim of this study was to investigate the impact of simulated OAE, through the alteration of pH, on phytoplankton representative of the spring and fall blooms in near-shore, temperate waters. The potential impacts were assessed through 1) an analysis of prior studies investigating the effects of progressively elevated pH on phytoplankton growth rates, and 2) by experimentally assessing the potential impact of elevated pH on the viability and growth rates of two representative near-shore species."

**Introduction**
In general: many references are missing throughout the whole introduction. Just a few examples: at line 36 after "drawdown"; at line 37 after "dominates". There's a huge literature to cite in the whole introduction that is completely missing.

Please see the response to the final Major Comment, and the below specific comments.

1. Lines 22-27 The first lines are out of topic. It seems more like an introduction to a thesis than an article. I suggest to find another way to introduce the study. This paragraph seems non-correlated to the next one.

    The reviewer is correct to infer that the manuscript arose from a MS thesis. Suggested rewording: "Climate change has become one of the most pressing problems facing us as a society, with atmospheric carbon dioxide ($CO_2$) concentrations steadily increasing over the past 250 years (Dlugokencky and Tans, 2018). This led to the signing of the Paris Agreement in 2015, with the agreed upon goal to keep the global average increase in temperature below 2 °C (*United Nations Framework Convention on Climate Change*, 2015). It is widely acknowledged, however, that reducing emissions will not be enough to meet this goal and carbon dioxide removal (CDR) will be needed. In fact, many of the IPCC scenarios that comply with the Paris Agreement regulations require as much as 10-20 Gt of $CO_2$ removal per year (Honegger and Reiner, 2018). To achieve this ambitious removal target Negative Emissions Technologies (NETs) will be needed."

2. Line 24: Galland et al., 2012: I guess many more references can be added here.

    Please see the response to Introduction comment 1 above.

3. Line 29: I disagree. NETs are not developed to combat rising atmCO2. See also recent literature on the need to reduce CO2 emission and on the minor role of CDR in this context. Please delete the first part of this sentence or try to put it in a different context.

   Suggest deleting this sentence.

4. Line 31: here and through the text take care about wording. Is OAE going to restore the pH and the carbonate system to their natural state? Why then study the impact of high pH if you consider OAE as a "restoring process"? This is a provocative question. I think this way of summarising the process is incorrect.

   Suggested rewording:
   "Ocean Alkalinity Enhancement (OAE) is one promising NET that involves anthropogenically raising the alkalinity, and as a result the pH, of a parcel of water causing the partial pressure of the $CO_2$ in that water to decrease. This change leads to either the uptake of $CO_2$ from the atmosphere or a reduction in the release of $CO_2$ from the ocean, depending on the initial air-sea gradient. Both scenarios result in a theoretical net reduction of atmospheric $CO_2$ and storage in the form of bicarbonate ($HCO_3^-$) and carbonate ($CO_3^{2-}$) ions in the ocean (Oschlies et al., 2023)."
   (Reviewer 1, Comment 5)

5. Line 33-38 there are different ways to apply OAE. Not all of them change the pH (i.e. increase the pH). In an equilibrated OAE for example, DIC is increasing while pH is rather stable. Please be cautious with this description and try to rephrase it. In the whole article, CO2 limitation is also barely mentioned.

   Suggested rewording:
   "There are currently several different methods of OAE in development, including mineral- and electrochemical-based methods, with deployment from vessels, through preexisting outfalls, or from placement on beaches. The focus of this study is the mineral-based approach from preexisting outfalls, implementation of which is likely to occur through addition of unequilibrated hydroxide minerals ($OH^-$) to the coastal surface ocean." (Reviewer 1, Comment 7)

6. Line 41: I would rephrase this part since at the moment we don't know how the regulations might change. There are ongoing projects that are trying to evaluate efficiency, efficacy and env. impact in the open ocean (ship). I would mind the words again and avoid saying: "almost certainly" I would suggest the authors refer to your possible case scenarios instead (i.e. land-based, coastal release of TA).

   Agreed. We suggest removing this paragraph in favour of the statement included in the above comment (Introduction 5, above).

7. Line 46: I would delete "especially" since the impact at different trophic levels is equally important

   Suggested rewording: substitute "including" for "especially".

8. Lines 49-64: some sentences are disconnected. Line 53-54: what does that mean? Why in this context do you think is important? References are missing and a big range of pH is put in the loop. This sentence it's a bit of a: "and then what?" sentence.

Suggested rewording for lines 53-56:
"The reason for the low concentration of dissolved $CO_2$ usable by phytoplankton is well illustrated in a Bjerrum plot (*e.g.*, Zeebe and Wolf-Gladrow, 2001) which shows bicarbonate's dominance of the inorganic carbon species at pH values of 6 – 9. Although $CO_2$ is the substrate for Rubisco, the prevalence of bicarbonate underlies a strong selective pressure among phytoplankton for the ability to utilize $CO_2$ in a carbon concentrating mechanisms (CCM). This is a trait observed across taxonomic groups (Colman et al., 2002; Nimer et al., 1997; Beardall et al., 2020). Different CCMs facilitate uptake of $CO_2$ by its active transport across the cell membrane and/or by uptake of $HCO_3^-$ through anion exchange, followed by its conversion to $CO_2$ by carbonic anhydrase (Coleman et al., 2002l Nimer et al., 1997; Beardall et al., 2020). Taxonomic differences in the energetic costs of different CCMs (Raven et al., 2014), in the pH optima of different forms of carbonic anhydrase (Idrees et al., 2017; Supuran, 2023), and in the specificity of different forms of Rubisco for $CO_2$ vs $O_2$ (Iñiguez et al., 2020) suggest that alkalization has the potential to alter community growth rates or to cause shifts in taxonomic structure within mixed assemblages."
In regard to the missing references, please see the response to Comment 9, below.

9. Lines 56-64: almost no references. They should be added.

Suggest adding the following references:
- Line 60 following "…gradient to function (Beardall and Raven, 2016)."
- Line 61 following "…or have carbonic anhydrase (Beardall et al., 1976; Raven and Hurd, 2012; Raven et al., 2014; Raven et al., 2017)."

10. Line 66: it is not only Hansen, 2002. There's a lot of literature on the response of phytoplankton to changes in pH in the optic of OA studies. Some of them tested also higher pH values!

Suggested rewording for: "(Hansen, 2002)" to "(see Supplement 1; Bach et al., 2015; Langer et al., 2006)."

11. Line 71: the aim of the study should be better clarified and underlined and not with an "en passant" sentence like this one.

Suggested rewording:
Delete the final sentence on lines 70-71 and modify lines 73-77 as "The study addressed potential impacts of OAE via the response of phytoplankton growth rates, viability, and photosynthetic competence via responses to elevated pH. First, published data were fitted to a model of growth to quantify the effect of progressively rising pH on the growth rates of a range of cultures phytoplankton. Second, the viability, growth rates, and photosynthetic competence (as $F_v/F_m$) were measured for two representative near-shore phytoplankton species, the diatom *Thalassiosira pseudonana* and the prymnesiophyte *Diacronema lutheri* (formerly *Pavlova lutheri*), following exposure to short- (10 minutes) and long-term (8 days) elevated pH."

**Literature Review & Data Digitization**

1.  Line 83: Since the literature review is not only based on Hansen 2022, this way of citing the studies is not correct. Please refer to the table in the supplementary. On top of that: why other studies that tested high pH on diatoms and or coccolithophores were excluded? The choice of the considered study is not clear to me.

    It is not immediately clear how the citation is inappropriate in this context as there is no specific citation on line 83. We proposed to clarify the criteria used to select the studies that were included in Section 2, as suggested below:
    A revised Methods section which would also include a better explanation of the rationale for picking the studies compared in Section 2: a literature survey in which our search terms included "phytoplankton AND (alkalinization OR "high pH")". Studies found in the search were gated to include those in which cultures were maintained under pH drift conditions, with culture medium ensuring that the media would have DIC:DIN below the Redfield Ratio, and with time-series (growth curves) of phytoplankton abundance and pH. (Reviewer 1, Comment 16)

**Examining the impact of prolonged, elevated pH on phytoplankton with and without DIC resupply**

1.  Line 174: The description of OAE is incorrect. First, there are different ways to apply OAE (equilibrated and non-equilibrated). DIC in a non-equilibrated approach is rather stable at the very beginning.

    Suggested rewording:
    "Most of the studies analyzed in Section 1 did not permit $CO_2$ in-gassing, as they were conducted using a closed-bottle, batch-culturing method. However, for OAE to successfully function as a NET, in-gassing of $CO_2$ is required to increase the DIC pool." (Reviewer 1, Comment 13)

2.  Line 192-194: This is more a method than a description of the results.

    These details can be moved to a proposed revised Methods section as described in response to the first Major Comment.

3.  It's not clear what this paragraph is about.

    We are unsure to which paragraph the reviewer is referencing in this comment.

4.  I don't know if this is the right paragraph but the way pH and the other carbonate chemistry parameters are not clear to me. What did you measure? DIC or TA on top of pH? And how?

    For this experiment, DIC and pH were measured, and the remaining carbonate chemistry parameters were calculated using CO2SYS (Lewis and Wallace, 1998). This would be stated explicitly in the proposed revision of the Methods.

5.  Lines 195-196: I strongly disagree with this statement. I would tone it down to the specific case that you are considering.

Suggested rewording:
"These results demonstrate that the test organism, the biological responses to alkalization depend in large part on $CO_2$ in-gassing, so the response to mineral-based OAE and subsequent in-gassing could not necessarily be inferred from pH drift experiments."

**Assessing the effects of short- and long-term alkalization on viability, growth, and photosynthetic competence in two coastal phytoplankton**

I found the whole of chapter 4 quite hard to follow. Methods and results are mixed in a way that makes the reading quite challenging. I strongly encouraged the authors to rethink the structure of the whole chapter 4 to make it easier to follow by the reader.

We propose to consolidate the methods used for the different sections of the paper into a single section. Please see the response to the first Major Comment.

**Discussion**

1. Line 305-306: the first sentence of the Discussion is out of topic as mentioned by the way by the authors. Why mention it as the first sentence MRV if it is unrelated to the aim of the study?

   Suggested rewording for the first two sentences (lines 305-307):
   "Where OAE is based on non-equilibrated alkalization, it will result in a rise in both alkalinity and pH. We have tested potential responses through a combination of data analysis and experimental manipulations, under conditions in which the pH increase is accompanied by DIC drawdown and in conditions in which the two are largely decoupled."

2. The very first sentence mentions the increase in pH. However, OAE could be applied in different ways (not only with the addition of NaOH) that could impact less the ecosystem. I mean the equilibrated approach that would not induce an increase in pH but in DIC. On top of that, even if the aim of the study is the impact of high pH on phytoplankton, what about CO2 limitation?

   Please see the response to Discussion Comment 1, above.

3. Line 308: a variation from 8.2-8.3 is it by the way in the natural range of variation in a marine context, meaning that phytoplankton is by definition able to cope with this (and even bigger) pH variations. Should we then be surprised that this shift is not making any impact on the studied species?

   Suggested rewording:
   "This result is not unsurprising considering the natural variations of pH in a marine system tend to fluctuate much more than this (Oberlander, 2023)."

4. Line 315: with cautions in which sense and why?

   Suggested rewording:
   "This suggests that evaluation of unequilibrated OAE in the context of pH-drift cultures should be completed with careful consideration of the associated changes in the DIC availability."

5.  In general, if OAE is applied in a non-equilibrated way as in the case of NaOH solutions, the authors should consider that equilibration will take time. Therefore, the perturbation of the carbonate chemistry could last relatively longer in the water

    We propose a comprehensive review of the manuscript to ensure that specific details regarding unequilibrated OAE are explained thoroughly.

6.  Line 310- 315: I strongly disagree. Equilibration could take time to happen. It is not such a fast process at least not in every oceanic/marine context. Ingassing could take longer, therefore, the pH-drift effect must be considered. I value the long/semi-continuous studies that should give us information on the long(er) response of the phytoplankton. But I disagree with the limited value of short(er) batch experiments that within some minutes/hours/days can give us a hint of how a species responds to the carbonate chemistry perturbation induced by increased TA

    Suggested rewording:
    "The mechanism and potential impacts of OAE allow for $CO_2$ invasion to restore the equilibrium concentration of $CO_2$ after the conversion of existing $CO_2$ into bicarbonate. Where the approach is not calibrated to raise alkalinity without perturbing pH, OAE will result is a rise in pH with adjustment of the DIC speciation towards bicarbonate and carbonate, followed by in-gassing of $CO_2$ to restore the equilibrium. If in-gassing is restricted, the rise in pH will be accompanied by a drawdown in DIC if there is an excess of photosynthesis versus respiration. Comparisons of alkalized cultures with high DIC (semi-continuous cultures or aerated batch cultures) and those in which there was a drawdown (sealed pH drift cultures) show that in some cases there is no difference in the resulting growth rates (the dinoflagellates *Ceratium furca* and *C. fusus*; Figure 2) but in others it is pronounced (the congeneric dinoflagellate *C. tripos* and the diatom *Thalassiosira pseudonana*; Figures 2 and 3). The potential effects of OAE are therefore likely to depend on the degree to which changes in pH and DIC are decoupled."

7.  Line 318: references missing after others

    Suggested rewording:
    "The analysis of prior studies of pH-dependant growth rates indicates that dinoflagellates are statistical outliers among the taxonomic groups and may be more sensitive to alkalization than most others (Table 1; Supplement 1)."

8.  Line 321: what is likely to be a high mean?

    Suggested rewording:
    "…summer assemblages in nearby Mariager Fjord when pH could reach a up to a value of 9 between the months of May and August (Hansen, 2002)."

9.  Line 322: compared to what? (i.e. which group?)

    Suggested rewording:
    "Their higher sensitivity to raise pH, when compared with the other species investigated in Table 1 and Figure 1, might also reflect fundamental…"

10. Line 336: this is the authors' idea since there are studies there that are now trying to apply OAE simulating the release from a ship in the open waters. I would suggest that the authors will use the opportunity to relate their study to the specific case of coastal water and/or basin OAE.

Suggested rewording:
"A last reason for caution in extrapolating the pH-drift responses lies with the most probable scenario for conducting unequilibrated OAE in coastal waters. Discharge would occur in the nearshore in this scenario, likely into strong lateral flow environments."

11. Line 355: London protocol see comment above

Suggested rewording removes reference to the London Protocol (see response to Discussion Comment 10, above).

12. Line 342-345: how accurate are these pH numbers? How were they calculated?

These values were calculated using the growth rate, corresponding initial pH, and Equation 2 from the manuscript. Suggested rewording:
"The threshold values for reductions in growth rate with chronic exposure were 8.59±0.059 and 8.68±0.199 for *T. pseudonana* and *D. lutheri*, respectively."

13. In the discussion, most of the studies on the impact of OAE on phytoplankton are not mentioned. The article is lacking in putting its results into a broader context considering studies like Gately et al, 2023.

Please see the response to the final Major Comment above.

---

## Author Response (AR1)

**RC1: Anonymous Referee #1**

*General Comments:*

There are three main conclusions made within this study. The first is that there were no significant impacts on the viability or growth rate of the two species assessed within the short term exposure to elevated alkalinity levels. This is to be expected as levels of OAE that would likely result in an impact within 10 minutes of exposure would be significantly higher than what is logistically possible or safe in terms of secondary precipitation. The second is that longer term exposure to elevated alkalinity resulted in a decrease in growth rate. This is an interesting finding as many other studies thus far have found little evidence to suggest that there will be significant impacts to the growth of phytoplankton as a result of OAE. The third is that within the current literature evidence suggests that approximately 50% of species could be significantly impacted by pH increases in line with those expected as a result of OAE. This provides an excellent summary of the current knowledge of pH impacts however it is unclear as to whether these impacts are a result of pH (H+ concentrations) or CO2 concentrations, of which the latter is expected to be the most important factor in regard to the ecological impacts of OAE on phytoplankton.

We would like to thank the reviewer for their time and effort put into their comments and useful feedback on how to improve this paper. Specific comments are addressed below.

*Major Comments:*

The manuscript does an excellent job in referencing and discussing the influence of elevated pH on phytoplankton growth and viability. However, citations and discussion around OAE specifically are currently lacking. Further I found it difficult to disentangle exactly how the experiments were run within this manuscript and strongly recommend that the methods/results section be revised so that readers may easily understand what the authors have accomplished here.

We thank the reviewer for this helpful comment and agree that referencing other studies on OAE needs to be greatly improved throughout the manuscript and that the structure of several of the sections (specifically sections 3 and 4) should be changed to better explain the methods and results. We have gone into further detail below in the specific comment response about how we plan to implement these changes, and what references we will be adding to the manuscript.

*Specific Comments:*

1. Lines 8 – 9: There are many methods for achieving OAE and only mineral based methods could be considered to "mimic" the natural weathering of alkali minerals. Furthermore, enhancing alkalinity does not lead to the sequestering of atmospheric CO2 but decreases oceanic CO2 concentrations allowing for atmospheric CO2 to be sequestered under the correct conditions. I appreciate that this is within the abstract and concise wording is necessary but would advise the authors to consider rewording this sentence.

   We agree that the wording here should be altered to more accurately describe the varied processes for OAE. We suggest changing this sentence to read "One proposed NET is Ocean Alkalinity Enhancement (OAE), in which anthropogenically raising the alkalinity, and subsequently the pH, through the consumption of protons leads to a decrease in the partial pressure of $CO_2$ in the water allowing for atmospheric $CO_2$ to then be sequestered (Oschlies et al., 2023)."

2. Line 11-12: This sentence is difficult to understand, consider rewording.

   Thank you, we have reworded to improve the clarity of the sentence. "… and by experimentally assessing the potential impact of elevated pH on the viability and growth rates of two representative near-shore species of phytoplankton."

3. Line 16 – please provide the actual number of days or range of days.

   The number of days (8) the long-term experiments ran for has now been included.

4. Lines 18-19: Within this manuscript the authors have only looked at two species from two different taxonomic groups. This statement is too broad for the results of the experiment conducted here.

   We agree that this sentence is perhaps too broad for the scope of this paper and suggest instead changing to the following. "These preliminary findings suggest that there will be little to no impact on the two studied representative species of near-shore, temperate water phytoplankton when OAE occurs in naturally flushed systems."

5. Line 31: I caution the authors against the use of the word "natural". All methods of CDR are anthropogenically motivated and furthermore depending on the method of OAE used it may be more or less similar to the natural weathering of alkali minerals (e.g. mineral based OAE vs electrochemical based OAE). Furthermore, it is highly unlikely that OAE would result in these changes as any increase in pH would ideally be negated by an influx of CO2, resulting in an increase in DIC beyond current and pre-industrial concentrations.

   We thank the reviewer for these comments and have reworded the sentence, and subsequent sentences with the word 'natural', to better represent the reality of OAE. Additionally, we have improved the clarity of the paragraph to better reflect that the scope of this work specifically focuses on mineral based OAE from land additions.
   "Ocean Alkalinity Enhancement (OAE) is one promising NET that involves anthropogenically raising the alkalinity, and as a result the pH, of a parcel of water causing the partial pressure of the $CO_2$ in that water to decrease. This change leads to either the uptake of $CO_2$ from the atmosphere or a reduction in the release of $CO_2$ from the ocean, depending on the initial air-sea gradient. Both scenarios result in a theoretical net reduction of atmospheric $CO_2$ and storage in the form of bicarbonate ($HCO_3^-$) and carbonate ($CO_3^{2-}$) ions in the ocean (Oschlies et al., 2023)."

6. Line 33: Ideally this would be the case however additions of alkalinity and subsequent in gassing of atmospheric CO2 will lead to increases of both carbonate and bicarbonate. This will depend on other variables, but without CO2 equilibration concentrations of carbonate would increase significantly more than bicarbonate.

   We believe this comment is addressed through the changes made in response to comment 5.

7. Line 34: "…would likely be…" . Also, it is not yet clear which method of OAE will be implemented at large scales. It is more likely that multiple methods will be implemented in different regions e.g. electrochemical in coastal regions and mineral based in pelagic regions.

The sentence has been changed to better reflect that the scope of this work focuses on mineral based OAE additions from land but acknowledges that there are other methods of OAE in development.

"There are currently several different methods of OAE in development, including mineral- and electrochemical-based, along with various deployment options, such as via vessels, through preexisting outfalls, or direct placement on beaches. This study is specifically investigating the mineral-based approach from preexisting outfalls, the implementation of which is likely to occur through the addition of unequilibrated hydroxide minerals ($OH^-$) to the coastal surface ocean."

8. Line 36: Again, enhancing alkalinity does not necessarily result in bicarbonate formation. For a parcel of water with TA 2100 µmol/kg, DIC 2000 µmol/kg, T 15°C and salinity 35 an increase in alkalinity of 500 µmol/kg would result in changes of bicarbonate – 316.93 µmol/kg, carbonate + 340 µmol/kg. Also, such changes in carbonate chemistry do not necessarily lead to CO2 drawdown this is dependent on several other factors.

We have removed this sentence in favour of the revision suggested in the response to comment 5.

9. Lines 84 – 85: The authors discuss changes in pH throughout this section but was there additional criteria to ensure that manipulations of pH resulted in changes in DIC that would be expected of OAE (particularly for those articles not by "Hansen and colleagues")? For example, using varied additions of sodium carbonate and HCl one can achieve similar pH values but drastically different concentrations of TA and DIC. Given this and the expected ranges of pH in OAE one would expect CO2 and not pH to be the driving factor to influence phytoplankton.

Unfortunately, at the time of the literature review, this was not a criteria included since the majority of papers found focused on either OA or on the natural pH variations in Mariager Fjord in Denmark. We agree that OAE induced changes in $CO_2$ is an important driving factor on phytoplankton, however we argue that pH will be equally important, especially in the region immediately adjacent to the addition point. The purpose of this literature review was to focus more directly on the impacts of elevated pH in order to obtain a first order estimate of which species could be at a higher risk for negative impacts of OAE.

10. Figure 1: The y axis of figure 1a/b are very hard to discern as they overlap and it is unclear which axis label is referring to which set of units on the axis.

More space has been added between Figure 1a and 1b to make the figures more accessible to the reader.

11. Figure 1d: It is very difficult to see the median value for the 90% µ reduction. Consider changing the colour of this line and/or adjusting the alpha values in this plot.

The median lines have been made longer and the colour changed to a light grey to improve clarity.

12. Lines 149 – 148: Is this anticipated maximum pH value for OAE based on changes in pH at the point of alkalinity addition or after dilution within a region?

This pH is the anticipated maximum at the point of addition, and we have added in a sentence to clarify this. "For all of the species investigated (Figure 1), the median threshold pH is above 8.5, suggesting about 50% of the species would not be impact by the anticipated maximum pH increase associated with a mineral-based addition of OAE from a land-based point source."

13. Lines 174: I recommend the authors adjust the wording of this sentence as OAE does not change DIC concentrations initially so there can be no "resupply" of DIC. More accurately this would be an influx of CO2 increasing DIC beyond what it was prior to OAE.

    We thank the reviewer for pointing out this factual error and have corrected it to more accurately describe the DIC relationship to OAE.
    "Most prior work investigating the impacts of elevated pH on phytoplankton growth did not permit for $CO_2$ resupply, as they were conducted using a closed-bottle, batch culturing method. However for OAE to successfully function as a NET, an influx of $CO_2$ will be required which will subsequently increase the DIC pool."

14. Lines 190 -192: If these observations are made on the basis of the Metric multidimensional scaling plots, please state this.

    The sentence has been reworded to clearly indicate the observations are from the mmds plots.

15. Lines 195 – 196: This is a strong statement to be made from a relatively small comparison. Furthermore, there are various approaches to OAE that would not necessarily result in CO2 influxes similar to those seen here e.g. in areas where CO2 outgassing occurs, or where OAE is added in an equilibrated form. Could the authors provide some references to support this statement?

    We agree with the authors concerns regarding this statement and have changed the word to better reflect the specific scenario of the experiment being discussed.
    "These results suggest that *T. pseudonana*'s response to unequilibrated mineral-based OAE cannot necessarily be inferred from pH drift experiments and further experiments are needed."

16. Lines 209 – 213: Please make it clear to the reader at the start of this section whether the cultures were grown in filtered seawater with the addition of nutrients and trace metals? Also, it would be beneficial to disclose the exact location of water collection i.e. xxx kms offshore via boat or pump. In addition, it is stated that "…fresh media in mid-exponential phase…" is this for the experimental cultures or the maintenance of culture pre experiment?

    We have added the word 'filtered' on line 210 before "seawater medium" to specify that the seawater used for the media is filtered.
    We describe the nutrients and trace metals on lines 209-211 by stating that the seawater medium was either f/2 or L1 and including the appropriate references. From our understanding of this comment, you are suggesting that this information be moved to the paragraph before, however we argue that it should remain where it is. The current position allows for the explanation regarding the choice of phytoplankton cultures followed by the culturing method, which we believe is the ideal order for reader comprehension.

Unfortunately, we do not have the information for the exact location of water collection. Lastly, we believe that renaming this section to '4.1 Culturing Techniques and Maintenance", as well as reorganizing the below sections will help to clarify that this description is of the maintenance of the cultures pre-experiment.

17. Line 250: The authors previously used a dark acclimation of 30 minutes, is there a reason for the change to 20 minutes?

    This was a grammatical error that has now been corrected. The dark acclimation time was 30 minutes here.

18. It is not clear how the transient exposure cultures discussed in section 4.3.2 are setup. Are these setup the same as those for the chronic exposure experiment? The authors also mention cultures measured after 1-2 hours, are these the same cultures or separate to the chronic and transient exposure cultures? This confusion may come from the layout in which the methods/results are presented. I am hesitant to suggest changes to this but it would be beneficial to the reader if the authors could be overly obvious in the explanation of how cultures were setup and whether cultures measured after 1-2 hours, several days and/or 10 minutes are the same cultures or separate cultures.

    We agree with the reviewer in that this section needs further clarification on the methods and plan to rework the sections as follows to explain the methodology and results in a more succinct manner. We believe that with the modification to the structural organization the differences between the setup of the transient and chronic experiments will be much easier to follow.
    > 4.1: Culturing Techniques and Maintenance
    > 4.2: Experimental Methods
    > 4.2.1: Modified Serial Dilution Culture – Most Probable Number Assay
    > 4.2.2: Chronic elevated alkalinity setup
    > 4.2.3: Transient elevated alkalinity setup
    > 4.3: Results of chronic vs. transient exposure to elevated alkalinity
    > 4.3.1: Response to chronic elevated alkalinity in *Thalassiosira pseudonana* and *Diacronema lutheri*
    > 4.3.2: Response to transient elevated alkalinity

19. Lines 294 - 296: The authors state "…there was no significant trend with the transient elevation in pH transient elevation in pH (Figure 4c)." However, the figure caption for figure 4 states "…measured after exposure to elevated alkalinity and pH for 1-2 hours…". Is the transient elevated alkalinity assessing the effect after 10 minutes as stated in line 286 or 1-2 hours?

    We understand how this statement comes across as confusing and think that altering the phrasing in addition to the above-mentioned changes in this sections structure will help to fix this issue. The cultures were exposed to the elevated alkalinity for 10 minutes before being diluted with the SDC-MPN method into untreated media. However, due to the time needed to finish the dilution series, and the constraint on personnel, the subsample of culture used to measure $F_v/F_m$ could not be read immediately following the 10 minute exposure and thus were read between 1-2 hours following exposure. While this time difference is not ideal, there was little we could do to prevent this situation.

20. Figures should be introduced in the order they appear in the text e.g. figure 5 is introduced in line 287 while figure 4c is introduced in line 296.

    We thank the reviewer for pointing out this placement error and have switched the order of the figures so that #4 is now #5 and vice versa. We believe that keeping all subfigures in figure 4 is beneficial for the reader to make visual comparisons in the data.

21. Lines 310-311: This sentence needs to be revised in line with comment 8.

    We agree and have revised the sentence as follows.
    "The mechanism of unequilibrated OAE in regions of the ocean where the initial air-sea gradient favours additional uptake of $CO_2$, allows for the invasion of $CO_2$ to occur and re-equilibrate the air-sea gradient. This additional $CO_2$ is then stored in the form of bicarbonate and carbonate (Oschlies et al., 2023)."

22. Lines 311 – 314: It would be beneficial if the authors could provide some reasoning behind why there are differences between these methods and how to abate these differences.

    A possible explanation for the differences could be physiological differences in the phytoplankton species and/or taxonomic differences. However, without having conducted further studies on these differences we don't feel it is appropriate to speculate on the topic in this manuscript.

23. Lines 354 – 356: This sentence is confusing particularly "…the dominant carbon species would be carbonate and calcite would begin precipitating…". Do the authors mean that carbonate is the dominant form of carbon and would precipitate into calcite? Traditionally when discussing precipitation omega values are provided, this would be beneficial here as many other articles discuss calcite precipitation in this form see Moras et al. (2022) and Schulz et al. (2023).

    The intention of this sentence was to state that carbonate is the dominant form of carbon and would precipitate into calcite at this pH. We agree that the wording here is slightly confusing and have reworded to get our point across more clearly, as well as including reference to crossing the omega thresholds. We believe that specific omega values are outside the scope of this work.
    "We note that at the pH in our experiments, the dominant carbon species would be carbonate and would be at a greater risk of precipitating into calcite, thus crossing the threshold for the saturation states for calcite ($\Omega_{calc}$) and/or aragonite ($\Omega_{arag}$; Schulz et al., 2023). This could lead to reductions in the availability of bicarbonate for the CCM and might account for the reductions in growth rates observed in Figure 4."

24. Lines 356 – 357: It would be beneficial to cite some of the many article's discussing the efficiency of OAE and impacts of calcite precipitation here.

    We agree and have included citations for Schulz et al. (2023), Moras et al. (2022), and Hartmann et al. (2023).

25. There have been numerous studies assessing the impact of OAE on natural assemblages of phytoplankton and cultures in recent years e.g. Gately et al. (2023) and Guo et al. (2023).

Although the authors discuss changes in pH there is no significant discussion surrounding the numerous papers on the impact of OAE on phytoplankton. The manuscript is significantly lacking in this regard and inclusion and discussion of such articles within the introduction and/or discussion would help to improve this manuscript.

We strongly agree with the reviewer on this point and plan to incorporate several of the recent publications studying the impact of OAE on phytoplankton throughout this manuscript. Namely, Gately et al. (2023), Iglesias-Rodríguez et al. (2023), Guo et al. (2023), Paul et al. (2024), Hutchins et al. (2023), Ferderer et al. (2022), and Subhas et al. (2022).

26. The authors discuss initial pH values, were there also end measurements for pH or prior to media refreshment? Volumes used were relatively low and cultures were grown into stationary phase, as such a significant change in pH over the duration of the experiment would be expected. I understand that during the SDC – MPN assay it would be expected that in gassing occurred maintaining pH at a relatively stable level, however this is not clear for the other experiments.

We did not collect measurements for pH at the end of the experiments conducted in tubes because, as you noted, there would be a significant change in pH over the experiment caused by the phytoplankton themselves. As this change is much more gradual and a result of the phytoplankton growth, we did not consider it as part of the experiment and focused on the initial change from our additions and the subsequent recovery.

27. Line 241-242: Was alkalinity measured or was it calculated? If it was calculated was this based on the additions of NaOH or another measured carbonate chemistry parameter? It would be beneficial to the manuscript to add measured or alkalinity values calculated from a second carbonate chemistry parameter, as many articles discuss OAE in terms of alkalinity increases (μmol/kg) and not pH increases.

We agree that clarification is needed here, and throughout, that the alkalinity values reported are calculated based on the measured pH and known DIC values and that alkalinity was not measured independently. We plan on completing a thorough review of the manuscript to clarify wherever alkalinity values are listed.

28. The quality of Figure S1 is extremely poor and makes it difficult to read. In addition, Figures S1 c,d appear to be concentrations of carbon species (CO2, HCO-3 and CO32-) not DIC as stated in the figure caption.

The quality of the figure has been improved and the caption corrected to accurately reflect the content of the graph.

29. It was difficult to distinguish between the experiments, variables measured, and methods used here. I strongly recommend the authors increase the clarity of the text discussing the experimental methods. For example, lines 209 – 210 "The cultures were grown in 40-mL volumes of f/2 (Guillard, 1975) or L1 (Guillard & Hargraves, 1993) seawater medium, and diluted into fresh media in mid-exponential phase in a laminar flow hood." Was this done for all experimental culture's transient, chronic and those in the SDC-MPN assay? Or was this only prior to the

experiment during the acclimation? Am I correct to understand that there are separate cultures with separate maintenance methods i.e. the SDC-MPN cultures, the chronic and transient? A strict section or table outlining the experiments conducted and their common and/or differences in methodology would be greatly beneficial, as it is currently difficult to tease apart exactly how this experiment was conducted.

We agree with the reviewer in that the current structure of Section 4 makes it difficult for the reader to determine the methods and variables for each of the experiments. We believe that the change in structure recommended in response to comment 18 will aid in the reader's comprehension of the methods/variables for each experiment. We would also like to thank the reviewer for suggesting a table outlining the experiments conducted and brief descriptions of the methodology. We agree that this would be beneficial in explaining the differences in experiments and plan to include such a table upon final revisions.

30. Lines 10-12: The authors introduce the analysis of prior studies here but fail to state any results from this within the abstract. This is a major component of the manuscript and requires a section detailing the results within the abstract.

We have added the following sentence to the abstract discussing the results of the analysis. "The analysis of prior studies yielded results suggesting wide variability in the response to elevated pH within and between taxonomic groups, with about 50% of species expected to not be impacted by unequilibrated mineral-based OAE."

*References:*

Ferderer, A., Chase, Z., Kennedy, F., Schulz, K. G., and Bach, L. T.: Assessing the influence of ocean alkalinity enhancement on a coastal phytoplankton community, Biogeosciences, 19, 5375–5399, https://doi.org/10.5194/bg-19-5375-2022, 2022.

Gately, J. A., Kim, S. M., Jin, B., Brzezinski, M. A., and Iglesias-Rodríguez, M. D.: Coccolithophores and diatoms resilient to ocean alkalinity enhancement: A glimpse of hope?, Science Advances, 9, eadg6066, doi:10.1126/sciadv.adg6066, 2023.

Guo, J. A., Strzepek, R. F., Swadling, K. M., Townsend, A. T., and Bach, L. T.: Influence of ocean alkalinity enhancement with olivine or steel slag on a coastal plankton community in Tasmania, Biogeosciences, 21, 2335–2354, https://doi.org/10.5194/bg-21-2335-2024, 2024.

Hartmann, J., Suitner, N., Lim, C., Schneider, J., Marín-Samper, L., Arístegui, J., Renforth, P., Taucher, J., and Riebesell, U.: Stability of alkalinity in ocean alkalinity enhancement (OAE) approaches – consequences for durability of CO2 storage, Biogeosciences, 20, 781–802, https://doi.org/10.5194/bg-20-781-2023, 2023.

Hutchins, D. A., Fu, F.-X., Yang, S.-C., John, S. G., Romaniello, S. J., Andrews, M. G., and Walworth, N. G.: Responses of globally important phytoplankton species to olivine dissolution products and implications for carbon dioxide removal via ocean alkalinity enhancement, Biogeosciences, 20, 4669–4682, https://doi.org/10.5194/bg-20-4669-2023, 2023.

Iglesias-Rodríguez, M. D., Rickaby, R. E. M., Singh, A., and Gately, J. A.: Laboratory experiments in ocean alkalinity enhancement research, in: Guide to Best Practices in Ocean Alkalinity Enhancement Research, edited by: Oschlies, A., Stevenson, A., Bach, L. T., Fennel, K., Rickaby, R. E. M., Satterfield, T., Webb, R., and Gattuso, J.-P., Copernicus Publications, State Planet, 2-oae2023, 5, https://doi.org/10.5194/sp-2-oae2023-5-2023, 2023.

Moras, C. A., Bach, L. T., Cyronak, T., Joannes-Boyau, R., and Schulz, K. G.: Ocean alkalinity enhancement – avoiding runaway CaCO3 precipitation during quick and hydrated lime dissolution, Biogeosciences, 19, 3537–3557, https://doi.org/10.5194/bg-19-3537-2022, 2022.

Oschlies, A., Bach, L. T., Rickaby, R. E. M., Satterfield, T., Webb, R., and Gattuso, J.-P.: Climate targets, carbon dioxide removal, and the potential role of ocean alkalinity enhancement, in: Guide to Best Practices in Ocean Alkalinity Enhancement Research, edited by: Oschlies, A., Stevenson, A., Bach, L. T., Fennel, K., Rickaby, R. E. M., Satterfield, T., Webb, R., and Gattuso, J.-P., Copernicus Publications, State Planet, 2-oae2023, 1, doi:10.5194/sp-2-oae2023-1-2023, 2023.

Paul, A. J., Haunost, M., Goldenberg, S. U., Hartmann, J., Sánchez, N., Schneider, J., Suitner, N., and Riebesell, U.: Ocean alkalinity enhancement in an open ocean ecosystem: Biogeochemical responses and carbon storage durability, EGUsphere [preprint], https://doi.org/10.5194/egusphere-2024-417, 2024.

Schulz, K. G., Bach, L. T., and Dickson, A. G.: Seawater carbonate chemistry considerations for ocean alkalinity enhancement research: theory, measurements, and calculations, in: Guide to Best Practices in Ocean Alkalinity Enhancement Research, edited by: Oschlies, A., Stevenson, A., Bach, L. T., Fennel, K., Rickaby, R. E. M., Satterfield, T., Webb, R., and Gattuso, J.-P., Copernicus Publications, State Planet, 2-oae2023, 2, https://doi.org/10.5194/sp-2-oae2023-2-2023, 2023.

Subhas, A. V., Marx, L., Reynolds, S., Flohr, A., Mawji, E. W., Brown, P. J., and Cael, B. B.: Microbial ecosystem responses to alkalinity enhancement in the North Atlantic Subtropical Gyre, Frontiers in Climate, 4, https://doi.org/10.3389/fclim.2022.784997, 2022.

**RC2: Anonymous Referee #2**

*General Comments:*

I read with pleasure the paper: "Assessing the impacts of simulated Ocean Alkalinity Enhancement on viability and growth of near-shore species of phytoplankton" by J. Oberlander and coauthors. This manuscript is of high interest for the assessment of OAE. The author digs into the available data set from the previous study that investigated the impact of increased pH on phytoplankton (outside the context of OAE) and performs culture experiments on two species, Thalassiosira pseudonana and Diacronema lutheri with two different approaches (batch culture and semi-continuous) to trace the (very) short impact of increase pH and the long(er) impact in the context of OAE.
That said, I think the manuscript needs significant revision before it is suitable for publication.

We would like to thank the reviewer for their time and effort put into their comments and useful feedback on how to improve this paper. Specific comments are addressed below.

*Major Comments:*

It was hard to follow the structure of the article. Some chapters and subchapters do not follow the standard structure of an article to the point that methodologies, statistics analyses and results are mixed.

We thank the reviewer for this comment and believe it has been sufficiently addressed below in response to the comments on Section 3 and 4.

The analysis of previous studies is almost at the beginning of the article, but it is not discussed adequately in the discussion and conclusion chapters. Some literature is omitted throughout the article. What about other studies made for ocean acidification that tested however higher pH values e.g. Bach et al., 2015? Or some of the Riebesell et al., studies on coccolithophore?

We thank the reviewer for noting that there is not sufficient discussion of the analysis of previous studies in the discussion and conclusion section of the manuscript, and we will work to correct this by including a paragraph outlining the conclusions from the analysis as well as discussion further explaining the connection with OAE.
In regard to the omitted literature in the analysis, we focused primarily on laboratory studies that included direct relationships between increasing pH and the concentration of cells or growth rate of the phytoplankton. The studies included in this analysis had this data easily available in their papers (*i.e.*, figures or reported values), and due to time constraints contacting authors to request access to their data that was not included in this format was not feasible. We agree that the work conducted by Bach, Riebesell, and many others on ocean acidification is valuable to understanding the potential impacts of OAE. We note that Bach et al. (2015) does include the pH and growth rate data for *C. pelagicus*, however because the focus of his study was $CO_2$ it did not appear in our literature search for prior studies investigating elevated pH where our search terms included "phytoplankton AND (alkalinization OR "high pH")". We believe that for the purposes of this manuscript, it would not be beneficial to redo the analysis of previous studies. However, we will note that there were several publications that were not included in the review, along with their citations, to indicate the plethora of studies that were unfortunately outside the scope of this analysis in hopes that readers will be compelled to look further into the literature.

In the discussions previous articles related to the response of phytoplankton to the perturbation induced by OAE are not mentioned (e.g. Gately et al., 2023) and in general many references are missing throughout the text.

We strongly agree with the reviewer on this point and plan to incorporate several of the recent publications studying the impact of OAE on phytoplankton throughout this manuscript. Namely, Gately et al. (2023), Iglesias-Rodríguez et al. (2023), Guo et al. (2023), Paul et al. (2024), Hutchins et al. (2023), Ferderer et al. (2022), and Subhas et al. (2022). As well as publications relating to the efficiency of OAE and the impacts of calcite precipitation: Schulz et al. (2023), Moras et al. (2022), and Hartmann et al. (2023).

*Specific Comments:*

**Abstract**
The abstract should be strongly revised. On top of some inaccuracies that I will report in the following lines, the reasons why studying the impact of high pH on phytoplankton is lost within the lines. Moreover, the study is based on two specific species that are important components of the phytoplankton community in specific marine contexts (i.e. near-shore, temperate waters). The final statement (lines 18-19) is therefore too general and misleading. The author should consider the findings of this study and not generalise their results to other taxonomic groups. The analysis of the prior study is lacking in the abstract even if it's an important part of this study. It should be mentioned.

We would like to thank the reviewer for the constructive comments on how to improve the clarity of the abstract. We feel that the changes listed in the response to comment 3 below address the concerns regarding why we studied the impact of high pH.
We have also reworded the final sentence (lines 18-19) in the abstract to better reflect findings of this specific study. "These preliminary findings suggest that there will be little to no impact on the two studied representative species of near-shore, temperate water phytoplankton when OAE occurs in naturally flushed systems."
Lastly, we have included a sentence discussing the analysis of the prior studies to make the abstract more representative of the paper as a whole. "The analysis of prior studies yielded results suggesting wide variability in the response to elevated pH within and between taxonomic groups, with about 50% of species expected to not be impacted by unequilibrated mineral-based OAE. However, flushing of the system (i.e. dilution) is needed to accurately represent the mechanics of OAE, which is why viability was assessed…"
More detailed responses are below.

1. Lines 7-8: technologies used twice.

   This has been corrected and the first use of technologies changed to tools.

2. Line 9: OAE is not only mimicking but is enhancing/accelerating the process. Moreover, it should be made more explicit why CO2 is ultimately sequestered from the atmosphere. The link is missing.

   We agree that this sentence should be reworded to more accurately describe the process of OAE and explain the link with $CO_2$ sequestration. It has been changed to: "One proposed NET is

Ocean Alkalinity Enhancement (OAE), in which anthropogenically raising the alkalinity and subsequently the pH through the consumption of protons, leads to a decrease in the partial pressure of $CO_2$ in the water allowing for atmospheric $CO_2$ to be sequestered (Oschlies et al., 2023)."

3. Lines 10-11: As mentioned before, the aim of the study and the need to understand the impact of increased pH on primary producers should be better expressed.

We have added a sentence to the abstract to better explain the aim of the study and the reasoning for examining the impact of increased pH, as well as improving the clarity of the sentence in lines 10-11.
"The aim of this study was to investigate the impact of simulated OAE, through the alteration of pH, on phytoplankton representative of the spring and fall blooms in near-shore, temperate waters. The potential impacts were assessed through 1) an analysis of prior studies investigating the effects of elevated pH on phytoplankton growth rates, and 2) by experimentally assessing the potential impact of elevated pH on the viability and growth rates of two representative near-shore species of phytoplankton."

**Introduction**
In general: many references are missing throughout the whole introduction. Just a few examples: at line 36 after "drawdown"; at line 37 after "dominates". There's a huge literature to cite in the whole introduction that is completely missing.

We agree and have added literature as listed below.

1. Lines 22-27 The first lines are out of topic. It seems more like an introduction to a thesis than an article. I suggest to find another way to introduce the study. This paragraph seems non-correlated to the next one.

We thank the author for this comment and have revised the paragraph as follows.
"Climate change has become one of the most pressing problems facing us as a society, with atmospheric carbon dioxide ($CO_2$) concentrations steadily increasing over the past 250 years (Dlugokencky and Tans, 2018). This led to the signing of the Paris Agreement in 2015, with the agreed upon goal to keep the global average increase in temperature below 2 °C (*United Nations Framework Convention on Climate Change*, 2015). It is widely acknowledged, however, that reducing emissions will not be enough to meet this goal and carbon dioxide removal (CDR) will be needed. In fact, many of the IPCC scenarios that comply with the Paris Agreement regulations require as much as 10-20 Gt of $CO_2$ removal per year (Honegger and Reiner, 2018). To achieve this ambitious removal target Negative Emissions Technologies (NETs) will be needed."

2. Line 24: Galland et al., 2012: I guess many more references can be added here.

The changes made in response to Introduction comment 1 above resolves this issue.

3. Line 29: I disagree. NETs are not developed to combat rising atmCO2. See also recent literature on the need to reduce CO2 emission and on the minor role of CDR in this context. Please delete the first part of this sentence or try to put it in a different context.

This sentence has been removed.

4. Line 31: here and through the text take care about wording. Is OAE going to restore the pH and the carbonate system to their natural state? Why then study the impact of high pH if you consider OAE as a "restoring process"? This is a provocative question. I think this way of summarising the process is incorrect.

   This sentence has been removed in favour of the following, more precisely worded statement. "Ocean Alkalinity Enhancement (OAE) is one promising NET that involves anthropogenically raising the alkalinity, and as a result the pH, of a parcel of water causing the partial pressure of the $CO_2$ in that water to decrease. This change leads to either the uptake of $CO_2$ from the atmosphere or a reduction in the release of $CO_2$ from the ocean, depending on the initial air-sea gradient. Both scenarios result in a theoretical net reduction of atmospheric $CO_2$ and storage in the form of bicarbonate ($HCO_3^-$) and carbonate ($CO_3^{2-}$) ions in the ocean (Oschlies et al., 2023).

5. Line 33-38 there are different ways to apply OAE. Not all of them change the pH (i.e. increase the pH). In an equilibrated OAE for example, DIC is increasing while pH is rather stable. Please be cautious with this description and try to rephrase it. In the whole article, CO2 limitation is also barely mentioned.

   The sentence has been changed to better reflect that the scope of this work focuses on unequilibrated mineral based OAE additions from land but acknowledges that there are other methods of OAE in development.
   "There are currently several different methods of OAE in development, including mineral- and electrochemical-based, along with various deployment options, such as via vessels, through preexisting outfalls, or direct placement on beaches. This study is specifically investigating the mineral-based approach from preexisting outfalls, the implementation of which is likely to occur through the addition of unequilibrated hydroxide minerals ($OH^-$) to the coastal surface ocean."

6. Line 41: I would rephrase this part since at the moment we don't know how the regulations might change. There are ongoing projects that are trying to evaluate efficiency, efficacy and env. impact in the open ocean (ship). I would mind the words again and avoid saying: "almost certainly" I would suggest the authors refer to your possible case scenarios instead (i.e. land-based, coastal release of TA).

   We agree with the reviewer in that there are several scenarios for OAE to be implemented, and that the regulations could change to allow for OAE to be implemented via vessels in the open ocean. Thus, we have removed this paragraph in favour of the statement included in response to Introduction comment 5.

7. Line 46: I would delete "especially" since the impact at different trophic levels is equally important

   We have deleted the word especially.

8. Lines 49-64: some sentences are disconnected. Line 53-54: what does that mean? Why in this context do you think is important? References are missing and a big range of pH is put in the loop. This sentence it's a bit of a: "and then what?" sentence.

We thank the reviewer for this comment and will work to increase the cohesion of the paragraphs. For line 53-54, the intention was to discuss the Bjerrum plot in which bicarbonate is the dominate carbon species between a pH of ~6-9. However, upon revision this sentence does not clearly make that point. It has been reworded as follows. "The reason for the low concentration of dissolved $CO_2$ usable by phytoplankton is well illustrated in a Bjerrum plot, where between the pH values of 6 and 9 bicarbonate is observed as the dominate carbon species. This means that many phytoplankton have to adapt to utilize bicarbonate as opposed to $CO_2$ (Colman et al., 2002; Nimer et al., 1997)."
In regard to the missing references, please see the response to the comment below.

9. Lines 56-64: almost no references. They should be added.

The following references were added:
   • Line 60 following "…gradient to function (Beardall and Raven, 2016)."
   • Line 61 following "…or have carbonic anhydrase (Beardall et al., 1976; Raven and Hurd, 2012; Raven et al., 2014; Raven et al., 2017)."

10. Line 66: it is not only Hansen, 2002. There's a lot of literature on the response of phytoplankton to changes in pH in the optic of OA studies. Some of them tested also higher pH values!

We agree with the reviewer that this is not the only reference to include here and have decided to change the "Hansen, 2002" reference to "see Supplement 1; Bach et al., 2015; Langer et al., 2006."

11. Line 71: the aim of the study should be better clarified and underlined and not with an "en passant" sentence like this one.

We have clarified the aim of the study as follows.
"The study aims to examine OAE's potential impacts on phytoplankton growth rates, viability, and photosynthetic competence via responses to elevated pH. First, published literature were collated and analysed to quantify the effect of elevated pH on the growth rates of a range of cultured phytoplankton. Second, the viability, growth rates, and photosynthetic competence (as $F_v/F_m$) were measured for two representative near-shore phytoplankton species, the diatom *Thalassiosira pseudonana* and the prymnesiophyte *Pavlova lutheri*, following exposure to short- and long-term elevated pH. Furthermore, the impact of prolonged, elevated pH on *T. pseudonana* both with and without DIC resupply was investigated."

**Literature Review & Data Digitization**
1. Line 83: Since the literature review is not only based on Hansen 2022, this way of citing the studies is not correct. Please refer to the table in the supplementary. On top of that: why other

studies that tested high pH on diatoms and or coccolithophores were excluded? The choice of the considered study is not clear to me.

We thank the reviewer for this comment and believe that the majority of this comment is addressed above in the response to the major comments but would like to reiterate the following as explanation for why the studies that we used were chosen. "We focused primarily on laboratory studies that included direct relationships between increasing pH and the concentration of cells or growth rate of the phytoplankton. The studies included in this analysis had this data easily available in their papers (*i.e.*, figures or reported values), and due to time constraints contacting authors to request access to their data that was not included in this format was not feasible."
We believe that including a sentence similar to the ones above will help to further explain why we used the studies we did for the analysis, as well as including a sentence stating that there is more literature available on the topic of OA and phytoplankton but could not be included here due to time constraints.

**Examining the impact of prolonged, elevated pH on phytoplankton with and without DIC resupply**
1. Line 174: The description of OAE is incorrect. First, there are different ways to apply OAE (equilibrated and non-equilibrated). DIC in a non-equilibrated approach is rather stable at the very beginning.

We agree with the reviewer that there are factual errors in this statement and have corrected it to more accurately describe the relationship between DIC and OAE.
"Most prior work investigating the impacts of elevated pH on phytoplankton growth did not permit for $CO_2$ resupply, as they were conducted using a closed-bottle, batch culturing method. However for OAE to successfully function as a NET, an influx of $CO_2$ will be required which will subsequently increase the DIC pool."

2. Line 192-194: This is more a method than a description of the results.

We agree that this sentence is more a description of methods rather than results. In the interest of improving the clarity of the manuscript for the reader, we plan on reorganizing the paragraphs from lines 180 – 196 so that there are a distinct paragraph for each the methods and the results. This will allow for us to create a more linear narrative in Section 3, with a paragraph each for introduction, methods, and results.

3. It's not clear what this paragraph is about.

We believe the reviewer is referring to the paragraph in lines 190-196, and based on the comment above we will be reworking Section 3 to follow a more linear narrative. With this change this paragraph will more clearly focus on the results from the experiment with and without DIC resupply.
"In the sealed (pH drift) incubation, there was a progressive draw-down of DIC to about 50% of the initial value and a rise in pH to $9.26 \pm 0.03$ in stationary phase (Days 4-6; Figure 3a). Based on the Metric multidimensional scaling plots, there is clear separation between the aerated and pH-drift cultures following the period of exponential growth (Days 0-1) in both the carbonate system parameters (Figure 3c) and the biological parameters (3d). Additionally, when comparing

the biological parameters to the carbonate parameters using BEST, the highest correlations were for combinations of $CO_2$ and 1 – 3 other variables (Spearman's R, 0.59 – 0.61; see Supplement 2). These results suggest that *T. pseudonana*'s response to unequilibrated mineral-based OAE cannot necessarily be inferred from pH drift experiments and further experiments are needed."

4. I don't know if this is the right paragraph but the way pH and the other carbonate chemistry parameters are not clear to me. What did you measure? DIC or TA on top of pH? And how?

   We agree that there is no clear explanation here of how the measurements were made, and believe that by adding a methods paragraph before line 180 this will be improved. For this experiment, DIC and pH were measured, and the remaining carbonate chemistry parameters were calculated using CO2SYS (Lewis and Wallace, 1998).

5. Lines 195-196: I strongly disagree with this statement. I would tone it down to the specific case that you are considering.

   We have reworded the sentence as follows to relate to the specific scenario discussed in this section. "These results suggest that *T. pseudonana*'s response to unequilibrated mineral-based OAE cannot necessarily be inferred from pH drift experiments and further experiments are needed."

**Assessing the effects of short- and long-term alkalization on viability, growth, and photosynthetic competence in two coastal phytoplankton**
I found the whole of chapter 4 quite hard to follow. Methods and results are mixed in a way that makes the reading quite challenging. I strongly encouraged the authors to rethink the structure of the whole chapter 4 to make it easier to follow by the reader.

We thank the reviewer for this comment and understand how the current structure of this section can be rather difficult to follow. We plan on restructuring this section in the following manner in order to more clearly define the methods and results from the experiments.
    4.1: Culturing Techniques and Maintenance
    4.2: Experimental Methods
    4.2.1: Modified Serial Dilution Culture – Most Probable Number Assay
    4.2.2: Chronic elevated alkalinity setup
    4.2.3: Transient elevated alkalinity setup
    4.3: Results of chronic vs. transient exposure to elevated alkalinity
    4.3.1: Response to chronic elevated alkalinity in *Thalassiosira pseudonana* and *Diacronema lutheri*
    4.3.2: Response to transient elevated alkalinity

**Discussion**
1. Line 305-306: the first sentence of the Discussion is out of topic as mentioned by the way by the authors. Why mention it as the first sentence MRV if it is unrelated to the aim of the study?

   We agree that the purpose of the sentence, which is we were only investigating the second aspect of public acceptance, perhaps does not come across quite as intended. In the interest of improving clarity we have removed the first two sentences of the discussion (lines 305-307).

2. The very first sentence mentions the increase in pH. However, OAE could be applied in different ways (not only with the addition of NaOH) that could impact less the ecosystem. I mean the equilibrated approach that would not induce an increase in pH but in DIC. On top of that, even if the aim of the study is the impact of high pH on phytoplankton, what about CO2 limitation?

We have removed this sentence from the manuscript, as stated in response to the comment above.

3. Line 308: a variation from 8.2-8.3 is it by the way in the natural range of variation in a marine context, meaning that phytoplankton is by definition able to cope with this (and even bigger) pH variations. Should we then be surprised that this shift is not making any impact on the studied species?

We agree with the reviewers point that this pH range is in the natural range of variation for many marine systems, and that it is not surprising that this shift would not make much on an impact on the studied species. We think that adding in a statement following the sentence on line 308 will help to better clarify that these results are not surprising.
"This result is not unsurprising considering the natural variations of pH in a marine system tend to fluctuate much more than this (Oberlander, 2023)."

4. Line 315: with cautions in which sense and why?

The use of 'caution' here is perhaps not the best choice of word, and we agree that further elaboration is needed.
"This suggests that evaluation of unequilibrated OAE in the context of pH-drift cultures should be completed with careful consideration of the associated changes in the DIC availability."

5. In general, if OAE is applied in a non-equilibrated way as in the case of NaOH solutions, the authors should consider that equilibration will take time. Therefore, the perturbation of the carbonate chemistry could last relatively longer in the water

We agree with the reviewer that it will take time for the carbonate chemistry to return to equilibration. We plan to do a comprehensive review of the manuscript to ensure that specific details regarding unequilibrated OAE are explained thoroughly.

6. Line 310- 315: I strongly disagree. Equilibration could take time to happen. It is not such a fast process at least not in every oceanic/marine context. Ingassing could take longer, therefore, the pH-drift effect must be considered. I value the long/semi-continuous studies that should give us information on the long(er) response of the phytoplankton. But I disagree with the limited value of short(er) batch experiments that within some minutes/hours/days can give us a hint of how a species responds to the carbonate chemistry perturbation induced by increased TA

We thank the reviewer for this comment and have altered the sentence on lines 310-311 as follows: "The mechanism of unequilibrated OAE in regions of the ocean where the initial air-sea gradient favours additional uptake of $CO_2$, allows for the invasion of $CO_2$ to occur and re-equilibrate the air-sea gradient. This additional $CO_2$ is then stored in the form of bicarbonate and carbonate (Oschlies et al., 2023)."

7. Line 318: references missing after others

   *We have included a reference back to Table 1 and to the Supplement 1 for the full list of studies included in the literature review.*

8. Line 321: what is likely to be a high mean?

   *We thank the reviewer for pointing out that there is no number listed, this has now been included. "…summer assemblages in nearby Mariager Fjord when pH could reach a up to a value of 9 between the months of May and August (Hansen, 2002)."*

9. Line 322: compared to what? (i.e. which group?)

   *We have reworded this sentence to include reference to Table 1 and Figure 1 in this manuscript. "Their higher sensitivity to raise pH, when compared with the other species investigated in Table 1 and Figure 1, might also reflect fundamental…"*

10. Line 336: this is the authors' idea since there are studies there that are now trying to apply OAE simulating the release from a ship in the open waters. I would suggest that the authors will use the opportunity to relate their study to the specific case of coastal water and/or basin OAE.

    *We agree with the reviewer that a specific scenario should be described here as opposed to making a general statement about OAE.*
    *"A last reason for caution in extrapolating the pH-drift responses lies with the most probable scenario for conducting unequilibrated OAE in coastal waters. Discharge would occur in the nearshore in this scenario, likely into strong lateral flow environments."*

11. Line 355: London protocol see comment above

    *We have removed the reference to the London Protocol as it does not directly relate to coastal deployments of OAE.*

12. Line 342-345: how accurate are these pH numbers? How were they calculated?

    *We thank the author for noting that there is no error associated with the pH values, and we have included these now. These values were calculated using the growth rate, corresponding initial pH, and Equation 2 from the manuscript.*
    *"The threshold values for reductions in growth rate with chronic exposure were 8.59±0.059 and 8.68±0.199 for T. pseudonana and D. lutheri, respectively."*

13. In the discussion, most of the studies on the impact of OAE on phytoplankton are not mentioned. The article is lacking in putting its results into a broader context considering studies like Gately et al, 2023.

    *We agree that there are a significant number of references that should be included in the discussion and plan to include the following: Gately et al. (2023), Iglesias-Rodríguez et al.*

(2023), Guo et al. (2023), Paul et al. (2024), Hutchins et al. (2023), Ferderer et al. (2022), and Subhas et al. (2022).

*References:*

Bach, L. T., Riebesell, U., Gutowska, M. A., Fegerwisch, L., and Schulz, K. G.: A unifying concept of coccolithophore sensitivity to changing carbonate chemistry embedded in an ecological framework, Progress in Oceanography, 135, 125-138, https://doi.org/10.1016/j.pocean.2015.04.012, 2015.

Beardall, J., Mukerji, D., Glover, H. E., and Morris, I.: The path of carbon in photosynthesis by marine phytoplankton, Journal of Phycology, 12, 409-417, https://doi.org/10.1111/j.1529-8817.1976.tb02864.x, 1976.

Beardall, J., and Raven, J. A.: Carbon Acquisition by Microalgae, In: Borowitzka M, Beardall J, Raven JA, eds., The physiology of microalgae, Developments in Applied Phycology, Springer, 6, 89–99, https://doi.org/10.1007/978-3-319-24945-2_4, 2016.

Dlugokencky E. and Tans P.: Trends in atmospheric carbon dioxide, National Oceanic & Atmospheric Administration, Earth System research Laboratory (NOAA/ERSL), 2018.

Ferderer, A., Chase, Z., Kennedy, F., Schulz, K. G., and Bach, L. T.: Assessing the influence of ocean alkalinity enhancement on a coastal phytoplankton community, Biogeosciences, 19, 5375–5399, https://doi.org/10.5194/bg-19-5375-2022, 2022.

Gately, J. A., Kim, S. M., Jin, B., Brzezinski, M. A., and Iglesias-Rodríguez, M. D.: Coccolithophores and diatoms resilient to ocean alkalinity enhancement: A glimpse of hope?, Science Advances, 9, eadg6066, doi:10.1126/sciadv.adg6066, 2023.

Guo, J. A., Strzepek, R. F., Swadling, K. M., Townsend, A. T., and Bach, L. T.: Influence of ocean alkalinity enhancement with olivine or steel slag on a coastal plankton community in Tasmania, Biogeosciences, 21, 2335–2354, https://doi.org/10.5194/bg-21-2335-2024, 2024.

Hartmann, J., Suitner, N., Lim, C., Schneider, J., Marín-Samper, L., Arístegui, J., Renforth, P., Taucher, J., and Riebesell, U.: Stability of alkalinity in ocean alkalinity enhancement (OAE) approaches – consequences for durability of CO2 storage, Biogeosciences, 20, 781–802, https://doi.org/10.5194/bg-20-781-2023, 2023.

Honegger, M., and Reiner, D.; The political economy of negative emissions technologies: consequences for international policy design, Climate Policy, 18, 306-321, doi:10.1080/14693062.2017.1413322, 2018.

Hutchins, D. A., Fu, F.-X., Yang, S.-C., John, S. G., Romaniello, S. J., Andrews, M. G., and Walworth, N. G.: Responses of globally important phytoplankton species to olivine dissolution products and implications for carbon dioxide removal via ocean alkalinity enhancement, Biogeosciences, 20, 4669–4682, https://doi.org/10.5194/bg-20-4669-2023, 2023.

Iglesias-Rodríguez, M. D., Rickaby, R. E. M., Singh, A., and Gately, J. A.: Laboratory experiments in ocean alkalinity enhancement research, in: Guide to Best Practices in Ocean Alkalinity Enhancement

Research, edited by: Oschlies, A., Stevenson, A., Bach, L. T., Fennel, K., Rickaby, R. E. M., Satterfield, T., Webb, R., and Gattuso, J.-P., Copernicus Publications, State Planet, 2-oae2023, 5, https://doi.org/10.5194/sp-2-oae2023-5-2023, 2023.

Langer, G., M. Geisen, K.-H. Baumann, J. Kläs, U. Riebesell, S. Thoms, and Young, J. R.: Species-specific responses of calcifying algae to changing seawater carbonate chemistry, Geochem. Geophys. Geosyst., 7, Q09006, doi:10.1029/2005GC001227, 2006.

Moras, C. A., Bach, L. T., Cyronak, T., Joannes-Boyau, R., and Schulz, K. G.: Ocean alkalinity enhancement – avoiding runaway CaCO3 precipitation during quick and hydrated lime dissolution, Biogeosciences, 19, 3537–3557, https://doi.org/10.5194/bg-19-3537-2022, 2022.

Oberlander, J. L.: Assessing the Impacts of Simulated Ocean Alkalinity Enhancement on Viability and Growth of Near-Shore Species of Phytoplankton, MSc, Dalhousie University, http://hdl.handle.net/10222/82384, 2023.

Oschlies, A., Bach, L. T., Rickaby, R. E. M., Satterfield, T., Webb, R., and Gattuso, J.-P.: Climate targets, carbon dioxide removal, and the potential role of ocean alkalinity enhancement, in: Guide to Best Practices in Ocean Alkalinity Enhancement Research, edited by: Oschlies, A., Stevenson, A., Bach, L. T., Fennel, K., Rickaby, R. E. M., Satterfield, T., Webb, R., and Gattuso, J.-P., Copernicus Publications, State Planet, 2-oae2023, 1, doi:10.5194/sp-2-oae2023-1-2023, 2023.

Paul, A. J., Haunost, M., Goldenberg, S. U., Hartmann, J., Sánchez, N., Schneider, J., Suitner, N., and Riebesell, U.: Ocean alkalinity enhancement in an open ocean ecosystem: Biogeochemical responses and carbon storage durability, EGUsphere [preprint], https://doi.org/10.5194/egusphere-2024-417, 2024.

Raven, J. A., Beardall, J., and Giordano, M.: Energy costs of carbon dioxide concentrating mechanisms in aquatic organisms, Photosynth. Res., 121, 111-124, doi: 10.1007/s11120-013-9962-7, 2014.

Raven, J. A., Beardall, J., and Sánchez-Baracaldo P.: The possible evolution and future of CO2-concentrating mechanisms. J. Exp. Bot., 68, 3701-3716, doi:10.1093/jxb/erx110, 2017.

Raven, J. A., and Hurd, C. L.: Ecophysiology of photosynthesis in macroalgae, Photosynth Res, 113, 105-125, https://doi.org/10.1007/s11120-012-9768-z, 2012.

Schulz, K. G., Bach, L. T., and Dickson, A. G.: Seawater carbonate chemistry considerations for ocean alkalinity enhancement research: theory, measurements, and calculations, in: Guide to Best Practices in Ocean Alkalinity Enhancement Research, edited by: Oschlies, A., Stevenson, A., Bach, L. T., Fennel, K., Rickaby, R. E. M., Satterfield, T., Webb, R., and Gattuso, J.-P., Copernicus Publications, State Planet, 2-oae2023, 2, https://doi.org/10.5194/sp-2-oae2023-2-2023, 2023.

Subhas, A. V., Marx, L., Reynolds, S., Flohr, A., Mawji, E. W., Brown, P. J., and Cael, B. B.: Microbial ecosystem responses to alkalinity enhancement in the North Atlantic Subtropical Gyre, Frontiers in Climate, 4, https://doi.org/10.3389/fclim.2022.784997, 2022.

UNFCCC, "Adoption of the Paris agreement" (FCCC/CP/2015/L.9/Rev.1,2015).

**Additional Comments:**

The paper presents two major points in abstract: 1) exposure to elevated pH does not have significant impacts on the growth rates of phytoplankton within the short-term (10-minute) period; and 2) the growth rate decreases significantly with long-term (days) exposure to elevated pH. It is a noble research endeavor to provide an understanding of how growth rate of different types of phytoplankton would be affected by elevated pH. However, upon reading through the paper, I found inconsistencies regarding these major points, and the abstract does not encompass many points discussed throughout the paper related to the pH-drift experiment. Additionally, the paper's structure is challenging to follow due to a lack of clear separation between the literature review and laboratory experiments, as well as between methods and results sections. In addition, the figures were challenging to understand.

We would like to thank you for taking the time to read our manuscript and provide the helpful comments. We agree that there are revisions needed in the abstract and the structure of the paper in order to better describe the experiments conducted. We plan to revise the paper structure (specifically Sections 3 and 4) to have distinct introduction, method, and results sections.

Figure 1 illustrates the correlation between pH, growth rate, and cell concentration, asserting that half of the species are impacted by the maximum pH increase and indicating that the threshold pH is above 8.5. However, Figure 1-d depicts the threshold around 8.7. Overall, the figures are difficult to interpret due to missing grid lines and inconsistencies between the descriptions and figures.

We thank you for the suggestions on how to improve the figures and have taken them into consideration for easier reader comprehension. We would like to note that the text does state "the median threshold pH is above 8.5" and that this number is referenced specifically because it is the anticipated maximum pH increase associated with a mineral-based addition of OAE from a land-based point source. The text in the manuscript has been revised in order to make this point clearer.

Figures 3-c) and d) lack sufficient explanation and should be revised to improve comprehension.

We thank you for this comment and will be reworking this section to have a clearer introduction, methods, and results section. With this revision we believe that the explanation of the methods resulting in these figures will become clearer and improve comprehension.

The paper addresses an important research question and presents valuable experimental data. However, revisions are needed to improve clarity, consistency, and interpretation of results. With appropriate revisions, the paper has the potential to make a significant contribution.

We thank the reviewer for their comments to help improve this manuscript.

**Relevant changes made to manuscript:**

Abstract

1. Lines 7-8 reworded to:
   a. "In response, new tools are being developed…"
2. Lines 8-9 reworded to:
   a. "One proposed NET is Ocean Alkalinity Enhancement (OAE), in which artificially raising the alkalinity favours formation of bicarbonate from $CO_2$, leading to a decrease in the partial pressure of $CO_2$ in the water, and a subsequent invasion of atmospheric resulting in net sequestration of atmospheric carbon."
3. Lines 10-11 added:
   a. "The aim of this study was to investigate the impact of simulated OAE, through the alteration of pH, on phytoplankton representative of the spring and fall blooms in near-shore, temperate waters."
4. Lines 11-12 reworded to:
   a. "The potential impacts of OAE were assessed through 1) an analysis of prior studies investigating the effects of elevated pH on phytoplankton growth rates and 2) by experimentally assessing the potential impact of short-term elevation of pH on the viability and subsequent growth rates of two representative near-shore species of phytoplankton."
5. Line 16 added "(8 days)"
6. Lines 18-19 deleted and added:
   a. "The analysis of prior studies indicates wide variability in the growth response to elevated pH within and between taxonomic groups, with about 50% of species expected to not be impacted by pH increase expected unequilibrated mineral-based OAE. To the extent that the growth responses reflect (largely unreported) parallel reductions in DIC availability, the susceptibility may be reduced for OAE in which CO2 in-gassing is not prevented."

Introduction

7. Lines 22-27 changed to:
   a. "Climate change has become one of the most pressing problems facing us as a society, with atmospheric carbon dioxide ($CO_2$) concentrations steadily increasing over the past 250 years (Dlugokencky and Tans, 2018). This led to the signing of the Paris Agreement in 2015, with the agreed upon goal to keep the global average increase in temperature below 2 °C (*United Nations Framework Convention on Climate Change*, 2015). It is widely acknowledged, however, that reducing emissions will not be enough to meet this goal and carbon dioxide removal (CDR) will be needed. In fact, many of the IPCC scenarios that comply with the Paris Agreement regulations require as much as 10-20 Gt of $CO_2$ removal per year (Honegger and Reiner, 2018). To achieve this ambitious removal target Negative Emissions Technologies (NETs) will be needed."
8. Line 29 deleted
9. Line 31 reworded to:
   a. "Ocean Alkalinity Enhancement (OAE) is one promising NET that involves anthropogenically raising the alkalinity of a parcel of water causing the partial pressure of the $CO_2$ in that water to decrease. This change leads to either in-gassing of $CO_2$ from the atmosphere or a reduction in out-gassing of $CO_2$ from the ocean, depending on the initial

air-sea gradient. Both scenarios result in a theoretical net reduction of atmospheric $CO_2$ through storage in the form of bicarbonate ($HCO_3^-$) and carbonate ($CO_3^{2-}$) ions in the ocean (Oschlies et al., 2023)."

10. Line 33 reworded to:
    a. "The additional carbon would be stored in the form bicarbonate ($HCO_3^-$) and carbonate ($CO_3^{2-}$) ions, with the former (which has a residence time of c. 1,000 years in the ocean) favoured after in-gassing of $CO_2$."

11. Lines 34-39 deleted and changed to:
    a. "There are currently several different methods of OAE in development, including mineral- and electrochemical-based methods, with deployment from vessels, through preexisting outfalls, or from placement on beaches. The focus of this study is the mineral-based approach from preexisting outfalls, implementation of which is likely to occur through addition of unequilibrated hydroxide minerals (OH-) to the coastal surface ocean."

12. Lines 41-44 deleted

13. Lines 45-47 moved to end of previous paragraph

14. Lines 53-64 reworded to:
    a. "The reason for the low concentration of dissolved $CO_2$ usable by phytoplankton is well illustrated in a Bjerrum plot (*e.g.*, Zeebe and Wolf-Gladrow, 2001) which shows bicarbonate's dominance of the inorganic carbon species at pH values of 6 – 9. Although $CO_2$ is the substrate for Rubisco, the prevalence of $HCO_3^-$ underlies a strong selective pressure among phytoplankton for the ability to utilize $CO_2$ in a carbon concentrating mechanisms (CCM). This is a trait observed across taxonomic groups (Colman et al., 2002; Nimer et al., 1997; Beardall et al., 2020). Different CCMs facilitate uptake of $CO_2$ by its active transport across the cell membrane and/or by uptake of $HCO_3^-$ through anion exchange, followed by its conversion to $CO_2$ by carbonic anhydrase (Coleman et al., 2002l Nimer et al., 1997; Beardall et al., 2020). Taxonomic differences in the energetic costs of different CCMs (Raven et al., 2014), in the pH optima of different forms of carbonic anhydrase (Idrees et al., 2017; Supuran, 2023), and in the specificity of different forms of Rubisco for $CO_2$ vs $O_2$ (Iñiguez et al., 2020) suggest that alkalization has the potential to alter community growth rates or to cause shifts in taxonomic structure within mixed assemblages."

15. Line 66 citation changed to:
    a. (see Supplement 1; Bach et al., 2025; Langer et al., 2006)

16. Lines 70-71 deleted

17. Lines 73-77 reworded to:
    a. "This study addressed potential impacts of OAE via the response of phytoplankton growth rates, viability, and photosynthetic competence via responses to elevated pH. First, published data were fitted to a model of growth to quantify the effect of progressively rising pH on the growth rates of a range of cultures phytoplankton. Second, the viability, growth rates, and photosynthetic competence (as $F_v/F_m$) were measured for two representative near-shore phytoplankton species, the diatom *Thalassiosira pseudonana* and the prymnesiophyte *Diacronema lutheri* (formerly *Pavlova lutheri*), following exposure to short- (10 minutes) and long-term (8 days) elevated pH."

Section 2

18. Changed section title from "A review of the pH dependence of growth rates" to "Methods"
    a. Section 2.3 "Culturing Techniques" moved from Section 4.1
    b. Section 2.4 added "Experimental Setup"
    c. Section 2.4.1 added "Comparison of aerated and pH-drift cultures of *Thalassiosira pseudonana*
    d. Section 2.4.2 "Modified Serial Dilution Culture – Most Probable Number Assay"" moved from Section 4.2 and retitled "Transient and chronic OAE effects on viability, growth, and photosynthetic competence"
19. Line 81 added:
    a. "The search terms included "phytoplankton AND "alkalinization OR "high pH"). Studies found in the search were gated to include those in which cultures were maintained under pH drift conditions, with culture medium ensuring that the media would have DIC:DIN below the Redfield Ratio, and with time-series (growth curves) of phytoplankton abundance and pH. Additional criteria included species/strains indicative of marine systems and the inclusion of irradiances and temperature in the methods."
20. Line 85 added:
    a. "...batch cultures without ventilation to replenish $CO_2$. It is critical to recognize that in pH-drift cultures such as these, both pH and DIC vary and that the biological response may be due to changes in $CO_2$ availability as much as, or more than, by pH (see Section 3, below)."
21. Lines 209-213 (now in Section 2.3) reworded to:
    a. "The cultures were maintained in 40-mL volumes of sterile-filtered f/2 (Guillard, 1975) or L1 (Guillard & Hargraves, 1993) seawater medium, and diluted into fresh media in mid-exponential phase in a laminar flow hood. The seawater was collected by pump about 100 m offshore from the National Research Council of Canada's Marine Institute at Ketch Harbour, NS, and tangential flow filtered on collection. It was refiltered through a 0.2-μm capsule filter (Cytiva Whatman Polycap Disposable Capsules: 75TC) and nutrient-enriched in autoclaved glassware or in sterile cell culture plates. Prior to experimentation, parent cultures were fully acclimated to the experimental growth conditions, continuous illumination at c. 190 μmol photons m$^{-2}$ s$^{-1}$ at a temperature of 18 ± 1°C, by maintaining them in balanced growth in semi-continuous culture (MacIntyre & Cullen, 2005)"
22. Added the following to Section 2.4.1 from Supplement 2:
    a. "Cultures of the diatom Thalassiosira pseudonana (CCMP 1335) in 40-ml glass tubes were maintained in balanced growth (i.e., in semicontinuous culture) in f/2 medium at 18 °C and under continuous illumination of 190 μmol photons m$^{-2}$ s$^{-1}$, provided by cool-white fluorescent bulbs (see Section 2.3 for more details). The cultures were used to inoculate 2-L volumes of f/2 medium (Guillard, 1973). One was sealed (pH-drift); the second (aerated) was bubbled with air that had been passed through an activated charcoal filter (MacIntyre and Cullen, 2005). Both were stirred with Teflon-coated magnetic stir bars.

    The two cultures were subsampled daily for analysis of DIC, pH, cell counts, and in vivo fluorescence. The DIC and pH were used to estimate the concentrations of $CO_2$, bicarbonate, and carbonate, using CO2SYS (Lewis and Wallace, 1998). Cell counts were performed with an Accuri C6 flow cytometer (BD, Franklin Lakes, NJ, USA), as described by MacIntyre et al. (2018). Fluorescence was measured with a FIRe

fluorometer (Satlantic, Halifax, NS, Canada), using fits of the induction curve to estimate the quantum yield of PSII electron transport ($F_v/F_m$) and the photosynthetic cross section ($\sigma$). See Section 2.3 for more details."

23. Lines 192-194 moved from Section 3 to Section 2.4.1:
    a. "The variation in the biological parameters was compared to the abiotic parameters using BEST, an iterative test based on correlations between a matrix of pairwise similarity coefficients based on the biotic data, against similar matrices of all possible combinations of 1-5 abiotic parameters."
24. Added table at end of Section 2.4.1 describing each of the experiments in this study
25. Line 250 (now in Section 2.4.2) typographical error corrected (20 minutes to 30 minutes)
26. Section 2.4.2 reworded to more clearly describe the methods used for the chronic and transient experiments.
    a. Sentences from Lines 255-272 moved to this section.

**Section 3**

27. Changed section title from "Examining the impact of prolonged, elevated pH on phytoplankton with and without DIC resupply" to "Results"
    a. Section 2.3 changed to 3.1 titled "A review of the pH dependence of growth rates"
28. Figure 1 altered by:
    a. Increased the space between figures a and b to create a clearer separation of the axes.
    b. Increased the size and changed the colour of the median line in d.
29. Changed Section 3 to 3.2
30. Changed Section 4 to 3.3
    a. Subsection numbering was also changed accordingly
31. Lines 148-149 reworded to:
    a. "For all of the species investigated (Figure 1), the median threshold pH is above 8.5, suggesting about 50% of the species would not be impacted by the anticipated maximum pH increase at the point of addition anticipated for a mineral-based addition of OAE from a land-based point source."
32. Line 174 reworded to:
    a. "Most of the studies analyzed in Section 1 did not permit $CO_2$ in-gassing, as they were conducted using a closed-bottle, batch-culturing method. However, for OAE to successfully function as a NET, in-gassing of $CO_2$ is required to increase the DIC pool."
33. Lines 190-192 reworded to:
    a. "There is clear separation in the mMDS (Figure 3) between the aerated and pH-drift cultures following the period of exponential growth (Days 0-1), both in the carbonate system parameters (Figure 3c) and in the biological parameters describing abundance and physiological status (3d)".
34. Lines 192-194 moved to Section 2.4
35. Line 195-196 reworded to:
    a. "These results demonstrate that for the test organism, the biological responses to alkalization depend in large part on $CO_2$ in-gassing, so the response to mineral-based OAE and subsequent in-gassing could not necessarily be inferred from pH drift experiments."
36. Figure 4 figure caption reworded to:
    a. "…Measurements of (c) the quantum yield of PSII electron transport, $F_v/F_m$, a measure of the proportion of functional reaction centres, and (d) the maximum fluorescence-based

specific growth rate, μm, following 10-minute exposure to elevated alkalinity and pH in *T. pseudonana* and *D. lutheri*. Fluorescence was measured 1-2 hours following exposure. Estimates of maximum growth rates are based on fits of the growth curve in samples that were diluted to$10^{-3}$ in the SDC–MPN assay. There was no significant trend in the data for $F_v/F_m$ in *D. lutheri* nor for μm in either species. The dashed lines are the mean values. The reduction in $F_v/F_m$ in *T. pseudonana* was fit to Equation 2 (dashed line). The estimated threshold pH for reduced $F_v/F_m$ is 8.86 ± 0.24."

37. Figure 4 or Figure 5 order switched to appear in the order they are introduced in the text.
    a. Figure 4 (now 5) moved to end of Section 3.3.2

Section 4

38. Changed from "Assessing the effects of short- and long-term alkalization on viability, growth, and photosynthetic competence in two coastal phytoplankton" to "Discussion and Conclusions"
39. Line 310-311 reworded to:
    a. "The mechanism of unequilibrated OAE in regions of the ocean where the initial air-sea gradient favours in-gassing of $CO_2$, allows for the invasion of $CO_2$ to occur and re-equilibrate acress the air-sea interface. The additional $CO_2$ is then stored in the form of bicarbonate and carbonate (Oschlies et al., 2023)."
40. Lines 354-356 reworded to:
    a. "We note that at the pH range in our experiments, the dominant carbon species would be carbonate and would be at a greater risk of precipitating into calcite, thus crossing the threshold for the saturation states for calcite (Ωcalc) and/or aragonite (Ωarag; Schulz et al., 2023). This could lead to reductions in the availability of bicarbonate for the CCM and might account for the reductions in growth rates observed in Figure 5."
41. The following citations added adder lines 356-357:
    a. (Schluz et al., 2023; Mors et al., 2022; Hartmann et al., 2023)
42. Lines 305-307 reworded to:
    a. "Where OAE is based on non-equilibrated alkalization, it will result in a rise in both alkalinity and pH. We have tested potential responses through a combination of data analysis and experimental manipulations, under conditions in which the pH increase is accompanied by DIC drawdown and in conditions in which the two are largely decoupled."
43. Sentence added following line 308:
    a. "This result is not surprising considering the natural variations of pH in a marine system tend to fluctuate much more than this (Oberlander, 2023)."
44. Line 315 reworded to:
    a. "This suggests that evaluation of unequilibrated OAE in the context of pH-drift cultures should be completed with careful consideration of the associated changes in the DIC availability."
45. Lines 310-315 reworded to:
    a. "The mechanism and potential impacts of OAE allow for $CO_2$ invasion to restore the equilibrium concentration of $CO_2$ after the conversion of existing $CO_2$ into bicarbonate. Where the approach is not calibrated to raise alkalinity without perturbing pH, OAE will result is a rise in pH with adjustment of the DIC speciation towards bicarbonate and carbonate, followed by in-gassing of $CO_2$ to restore the equilibrium. If in-gassing is restricted, the rise in pH will be accompanied by a drawdown in DIC if there is an excess of photosynthesis versus respiration. Comparisons of alkalized cultures with high DIC

(semi-continuous cultures or aerated batch cultures) and those in which there was a drawdown (sealed pH drift cultures) show that in some cases there is no difference in the resulting growth rates (the dinoflagellates *Ceratium furca* and *C. fusus*; Figure 2) but in others it is pronounced (the congeneric dinoflagellate *C. tripos* and the diatom *Thalassiosira pseudonana*; Figures 2 and 3). The potential effects of OAE are therefore likely to depend on the degree to which changes in pH and DIC are decoupled."

46. Line 318 reference added:
    a. (Table 2; Supplement 1)
47. Line 321 reworded to:
    a. "…summer assemblages in nearby Mariager Fjord when pH could reach a up to a value of 9 between the months of May and August (Hansen, 2002)."
48. Line 322 reworded to:
    a. "Their higher sensitivity to raise pH, when compared with the other species investigated in Table 2 and Figure 1, might also reflect fundamental…"
49. Line 336 reworded to:
    a. "A last reason for caution in extrapolating the pH-drift responses lies with the most probable scenario for conducting unequilibrated OAE in coastal waters. Discharge would occur in the nearshore in this scenario, likely into strong lateral flow environments."
50. Line 343 added the calculated error:
    a. "$8.59 \pm 0.059$ and $8.68 \pm 0.199$"
51. Addition of paragraph discussing recently published literature on OAE effects on phytoplankton
    a. Gately et al. (2023), Guo et al. (2024), Ferderer et al. (2022), and Paul et al. (2024)

---

## Author Response (AR2)

**RC1: Anonymous Referee #1**

We thank the reviewers for their constructive, detailed, and very helpful analyses of the draft manuscript. Responses to their comments are given below.

*Comments:*

Abstract:

1. Lines 6-8: As I mentioned in the previous version, I think it's essential not to confound the role of NETs versus decarbonisation. NETs will be only a limited part of what we should do to reduce atm CO2. Please rephrase this sentence to avoid any potential misunderstanding. I appreciate that this concept is now well rephrased in the Introduction, though.
   The sentence has been reworded as follows:
   "In response, new tools are being developed to remove carbon from the atmosphere using negative emission technologies (NETs), in addition to reducing man-made emissions."

2. Lines 13-14: rephrase as "short-term (10 minutes) and longer-term (8 days)" to put already the time into context. Otherwise, there is a lack of information for the reader.
   The sentence has been rephrased as suggested by the reviewer.

3. Line 22: expected x 2. Change term.
   The second use of 'expected' has been changed to 'anticipated.'

Introduction:

4. Line 37: air-sea gradient: citation needed
   The citation for Oschlies et al., 2023 has been added following this sentence

5. Line 68: "without reducing their growth rate to zero (Berge et al., 2010; Hansen, 2002)." Can you make it more explicit that the growth rate is reduced without going to zero? The flow of the text would work better with the subsequent sentence.
   The sentence has been reworded as follows:
   "… that are able to maintain growth rates that, though lower than their maximum are still positive, at pH values above nine and below seven (Berge et al., 2010; Hansen, 2002)."

Methods:

6. Line 136: Diacronema lutheri (formerly Pavlova lutheri): check the text. In Figure 4, for example, you use P. lutheri instead of D. lutheri.
   The figure has been corrected, and the text has been checked to correct all instances of *P. lutheri* to *D. lutheri*.

7. Table 1 is very helpful. Standard errors should be included in the pH values. On top of that, when the pHs were measured? Just after the manipulation? For example, in the Effect of Chronic Alkalization on Growth Rate? Can you provide this information?
   Rather than include too much information in Table 1, thus causing confusion, we have elected to add sentences to lines 195 and 197 describing when the pH measurements were collected for both the transient and chronic experiments.

Line 195: "The first was used immediately for measurement of pH and the serial dilution."
Line 197: "The pH was measured immediately following manipulation with the NaOH."

Results:

8. Line 246: do you mean Figure 1d?
The reference has been corrected to Figure 1d.

9. Figure 2: I found this figure quite unclear. It either needs to be modified, or the caption should be made more explicit. Even after reading the corresponding paragraph (Lines 263-269 and the preceding 254-261), I still find it difficult to follow. Am I missing something in the text? I have re-read it several times, but I still don't understand, for example, what the three points refer to. Or: "The black lines represent the mean values" refer to exactly?
The caption for Figure 2 was changed to better describe the data.
"*Figure 2:* Comparison of the pH dependencies of growth in the dinoflagellates *Ceratium furca*, *C. fusus*, and *C. tripos* under two different growth conditions. Cultures were maintained at a constant pH in semi-continuous culture (Constant pH) and were kept in batch cultures in which the pH was allowed to drift (pH Drift). For each comparison, the symbols are the estimates from fits to Eq. 2 of the threshold pH, above which the growth rate declines (pHTh); the pH at which growth rate is reduced by 10%; and the pH at which growth rate is reduced by 90%. The horizontal black lines are the mean values across species under each growth condition. Growth rates in *C. tripos* did not vary over the range of pH tested when pH was kept constant, although they declined over the same range in the pH-drift experiment. Consequently, there are 2 estimates for the Constant pH condition and 3 for the pH Drift condition."

10. Caption for Figure 4 or more in general in the text: do you have any explanation for the scatter trends in the growth of T. pseudonana in Figure 4d?
The most likely explanation for the variance in *T. pseudonana* is limited observations in the period of active growth reflected in the fits. The maximum growth rate was estimated from 3 data points.

Discussion:

11. Line 329: Use always one terminology. In most of the paper, you use "unequilibrated", so I recommend using this term.
This has been corrected throughout the text.

12. Line 332: The synthesis of studies from the literature suggests that significant long-term increases in pH to 8.2–8.3: what does "significant" mean?
The term 'significant' has been changed to 'prolonged'.

13. Line 333: what do you mean by: "However, the conclusion needs to be qualified in the context of OAE, as the experiments were conducted in pH-drift cultures in which CO2 replacement is prevented". Do you mean that the algae were experiencing some kind of CO2 limitation? I doubt this is the case with an increase in pH of 0.1. Could you please comment on that or rephrase or consider deleting this part since there's already a nice explanation after this sentence?
We have elected to delete this sentence to improve reader comprehension.

14. Lines 350-353: can you rephrase? When you talk about "nearby", it took until the end of the sentence that you are referring to the works by Hansen et al.
The citations have been moved to immediately follow the statement "They were the primary focus for these researcher's studies (Hansen, 2002;…)".

15. Line 368: "Discharge in the nearshore in this scenario, likely into strong lateral flow", the verb is missing.
The sentence has been rephrased to:
"In this scenario, discharge in the nearshore will likely be into strong lateral flow."

16. From line 373 onwards, the discussion requires revision. I have a few suggestions and comments:
    a.  Lines 373 -383: at times, it was unclear what the authors were referring to. For example, in lines 373-375: "There were reductions in growth rate at high pH in both", I don't see any of these reductions, especially in the transient approach. Am I wrong? Where are the reductions in growth in Figures 4 and 5, as stated in the text (looking at the standard deviations reported in Figure 5, too)?
    We have removed the sentence on lines 373 – 375, "There were reductions in growth rate at high pH in both."
    The following paragraph (referred to in comment 16b) discusses how there are clear reductions in growth rate in the chronic exposure experiments. We have added in reference to the corresponding figure (4b), which shows the decline in growth rate with increasing pH for both species tested. Figure 5 refers to the viability of the species, which was only tested for the transient exposure experiments.

    b.  The authors also mentioned Fv/Fm without specifying the species they are referring to, though I assume they refer to the strong reduction in Fv/Fm in T. pseudonana following the 10-minute exposure to elevated alkalinity at pH 9.09. From this point, the authors discuss the results from Gately et al., 2023. When comparing the data from this study with Gately et al., 2023, are you only considering T. pseudonana? Since Diacronema lutheri has a very stable Fv/Fm regardless of the pH.
    The statement "in either species" was added following "…photosynthetic efficiency" on Line 384 to make clear to the reader that there was no reduction in Fv/Fm for either species during the chronic exposure experiment.
    Line 388 was also amended to reflect this: "not statistically different form the controls, as they were in this study for both species (Figure 4b)."

    c.  But then, in the subsequent paragraph, from line 385, the authors discuss the different responses in terms of photosynthetic efficiency of T. pseudonana and Diacronema lutheri. I am confused. This part needs to be rewritten and corrected, looking at the data carefully.
    The previous paragraph (which is referred to in comment 16b) is discussing only the results from the chronic exposure experiments, as stated at the beginning of the paragraph. The following paragraph (this comment, 16c) is discussing the results of the transient exposure experiments (lines 391 – 392), and then comparing this with the lack of effect in the chronic experiments (lines 392 – 394).

    We believe that with the changes outlined in response to comment 16b and more explicit reference to the figures, this paragraph will no longer be confusing as written.

17. Lines from 392: the discussion starting at line 392 is also a bit out of context since the topic here is not the efficiency of OAE.

    We are unsure what the reviewing is referencing here, as the sentences following line 392 discuss how crossing the saturation thresholds for calcite and/or aragonite would be detrimental for the carbon concentrating mechanisms in phytoplankton. The next two sentences state that though this is not an ideal scenario from an OAE perspective, it could possibly occur through accidental discharge and thus the data is useful for understanding a 'worse-case scenario'.

18. Lines 398-407 are interesting since the authors discuss previous studies on OAE and the phytoplankton community responses. However, the authors lack the chance to compare the different approaches. Paul et al. 2004 describe an equilibrated OAE approach, while Federer et al., 2023 and Guo et al., 2024 similar to this study, use a non-equilibrated approach. This is an opportunity for the authors to provide a more comprehensive overview of the literature and place their results into this context. However, the differences in the carbonate chemistry perturbations of these different mesocosm and microcosm experiments must be explicitly described.

    While the comparison of the equilibrated vs unequilibrated is important, we do not feel that providing a comprehensive overview of this topic is within the scope of this paper. Instead, we have added the following to lines 406 and 408 to more explicitly describe the perturbation method used by each of the referenced authors.
    Line 406: "Both Guo et al. (2024) and Ferderer et al. (2022), who similarly to this study used an unequilibrated OAE approach…"
    Line 408: "However, Paul et al. (2024), who used an equilibrated OAE approach, …"

19. Last but not least, I suggest the authors read and incorporate (if that's the case) the papers by
    a. Nina Bednaršek et al., under discussion,
       https://egusphere.copernicus.org/preprints/2024/egusphere-2024-947/
    b. Faucher et al., under discussion,
       https://egusphere.copernicus.org/preprints/2024/egusphere-2024-2201/

    We thank the reviewer for suggesting these papers and have considered them carefully for addition to this manuscript.